# Spice-Derived Phenolic Compounds: Potential for Skin Cancer Prevention and Therapy

**DOI:** 10.3390/molecules28176251

**Published:** 2023-08-25

**Authors:** Janette Baloghová, Radka Michalková, Zuzana Baranová, Gabriela Mojžišová, Zuzana Fedáková, Ján Mojžiš

**Affiliations:** 1Department of Dermatovenerology, Faculty of Medicine, Pavol Jozef Šafárik University, 040 01 Košice, Slovakia; janette.baloghova@upjs.sk (J.B.); zuzana.baranova@upjs.sk (Z.B.); zuzana.fedakova@upjs.sk (Z.F.); 2Department of Pharmacology, Faculty of Medicine, Pavol Jozef Šafárik University, 040 01 Kosice, Slovakia; radka.michalkova@upjs.sk; 3Center of Clinical and Preclinical Research MEDIPARK, Faculty of Medicine, Pavol Jozef Šafárik University, 040 01 Kosice, Slovakia; gabriela.mojzisova@upjs.sk

**Keywords:** melanoma, non-melanoma skin cancer, spice, phenolic compounds, anticancer effect

## Abstract

Skin cancer is a condition characterized by the abnormal growth of skin cells, primarily caused by exposure to ultraviolet (UV) radiation from the sun or artificial sources like tanning beds. Different types of skin cancer include melanoma, basal cell carcinoma, and squamous cell carcinoma. Despite the advancements in targeted therapies, there is still a need for a safer, highly efficient approach to preventing and treating cutaneous malignancies. Spices have a rich history dating back thousands of years and are renowned for their ability to enhance the flavor, taste, and color of food. Derived from various plant parts like seeds, fruits, bark, roots, or flowers, spices are important culinary ingredients. However, their value extends beyond the culinary realm. Some spices contain bioactive compounds, including phenolic compounds, which are known for their significant biological effects. These compounds have attracted attention in scientific research due to their potential health benefits, including their possible role in disease prevention and treatment, such as cancer. This review focuses on examining the potential of spice-derived phenolic compounds as preventive or therapeutic agents for managing skin cancers. By compiling and analyzing the available knowledge, this review aims to provide insights that can guide future research in identifying new anticancer phytochemicals and uncovering additional mechanisms for combating skin cancer.

## 1. Introduction

Spices are defined as aromatic vegetable substances derived from various parts of plants, including seeds, bark, roots, fruits, and leaves. They have been used globally for centuries as a food ingredient to enhance flavor and extend the shelf life of food [1]. Besides their culinary uses, spices are also recognized for their medicinal properties and have been traditionally employed for treating various diseases and ailments. In recent years, there has been a notable increase in research exploring the potential health benefits of spices. Studies suggest that spices may provide protection against conditions such as cardiovascular disease, neurodegenerative disorders, chronic inflammation, obesity, or type 2 diabetes [2,3,4]. The wide range of health benefits associated with spices can be credited to their various active components, with phenolic compounds being the predominant phytochemicals responsible for these effects [5].

In the last few decades, phenolic compounds have been intensively studied as potential anticancer agents against various types of cancers including breast cancer [6,7,8,9,10,11]. Moreover, based on the scientific literature, there is strong evidence supporting the potential of phenolic compounds as promising agents in the battle against both melanoma and non-melanoma skin cancers.

This review article provides an overview of the research on spice-derived phenolic compounds and their effects on melanoma and non-melanoma skin cancers, encompassing *in vitro* and *in vivo* studies.

## 2. Skin Cancers

Skin cancers encompass a range of malignancies that can affect individuals worldwide. Skin cancer incidence and mortality rates vary depending on several factors, including geographical location, sun exposure, skin type, and preventive measures. Skin cancers are among the most commonly diagnosed cancers globally. Non-melanoma skin cancers (NMSC), including basal cell carcinoma (BCC) and squamous cell carcinoma (SCC), are more prevalent than melanoma [12]. On the other hand, melanoma, although less common than NMSC, is more aggressive and has a higher potential for metastasis. Over the past few decades, there has been a notable rise in the incidence of melanoma, particularly among fair-skinned populations with significant sun exposure [13]. However, studies indicate that there will be a projected decline in melanoma rates in the future [14].

**Basal cell carcinoma (BCC)** is derived from immature pluripotent cells of the lower layers of the epidermis. The etiology is multifactorial. Exposure to UV radiation is the most important cause, as BCC occurs predominantly in places exposed to the sun. Previous exposure to chemical carcinogens, ionizing radiation or phototherapy for skin diseases, chronic inflammatory changes, or trauma can also contribute to BCC. A higher incidence of BCC is observed in immunocompromised patients. Multipotent differentiation potential is reflected in great clinical diversity and morphology, including nodular—solid and cystic BCC, ulcerative, superficial, morphea-like sclerosing, keratotic, and pigment variant. The ulcerative BCC can lead to extremely extensive lesions. Metastases are extremely rare, and the morbidity associated with BCC is related to local tissue invasion and destruction [15,16,17].

**Actinic keratosis (AK)** is the proliferation of cytologically atypical keratinocytes. Nowadays AK is considered an early in situ squamous cell carcinoma. Age, the cumulative dose of UV radiation, outdoor activities, male gender, solar lentigines, and patient’s immunological condition are considered risk factors. Scaly or hyperkeratotic brown-red, brown-yellow macules, or papules with an erythematous base, usually less than 1 cm in diameter, in sun-exposed parts of the body. AKs can either rare spontaneously regress, they can remain stable without significant change, or progress to invasive squamous cell carcinoma [18,19,20].

**Squamous cell carcinoma (SCC)** represents the most aggressive type of non-melanoma skin cancer. Up to 60% of invasive SCCs arise from previous AK. Actinic keratosis, radiation (arsenic) keratosis, Bowen disease, Bowenoid papulosis, actinic cheilitis, and Queyrat erythroplasia are considered to be its in situ variants. The etiology is multifactorial and includes mainly chronic exposure to UV radiation, light phototype, long-term previous phototherapy for other dermatoses, professional exposure to X-ray radiation or radiation for internal malignancies, chemical carcinogens, chronic inflammatory skin changes, immunosuppression, and infection with HPV viruses. It grows *de novo* from a previous precancer. Initially, indurated, painless papules or macules of grayish, brownish-yellow, and reddish skin color may progress to nodular lesions with crusts and ulceration on the surface. The tumor grows quickly, disintegrates, and can outgrow soft tissues, cartilage, and bone. It can metastasize to regional lymph nodes, and later also to other organs. The prognosis depends on the location, size, and degree of tumor differentiation [20,21,22].

Early and correct diagnosis of NMSCs is a basic prerequisite for their successful treatment. The gold standard for the diagnosis of BCC, SCC, and AK is histopathological examination. The choice of individual therapeutic strategies depends on the anatomical location, the thickness of the tumor, the affected area of the skin surface, the histological type of the tumor, patient comorbidity, and the availability of the method. Surgical excision is still the predominant therapeutic strategy. Other treatment possibilities include curettage, electrodesiccation, cryotherapy, and radiotherapy. Only a few drugs meet the criteria for field treatment: 5% fluorouracil, 3% diclofenac in a gel with hyaluronic acid, photodynamic therapy, chemical peeling (destruction is achieved with 35% trichloroacetic acid, alpha-hydroxy acids, zinc chloride, and phenolic acid), retinoids (topical or systemic), and 5% imiquimod in cream [20,21,22,23,24,25].

**Melanoma** is a malignant skin tumor that arises from the malignant transformation of melanocytes anywhere in the human body. It is mainly localized in the skin, but it also often appears in the eye or mucous membranes. It belongs to the most insidious tumors due to its ability to quickly form metastases. One-third of melanomas arise from various pigment nevi, and two-thirds of melanomas arise from so-called “*de novo*”—without a pre-existing pigmented lesion on clinically normal skin. Melanoma is considered to be the human tumor with the greatest immunogenic response. The incidence of melanoma is increasing more than the incidence of any other malignancy [26,27].

The causes of the malignant transformation of a melanocyte into a melanoma cell are still unclear. The genetic predisposition of an individual with a positive family history of melanoma in combination with the influences of the external environment plays an important role in the development of melanoma. UV radiation is one of the best-documented risk factors for the development of melanomas—whether from existing dysplastic nevi or arising *de novo*. Other risk factors include burning the skin at a young age, phototherapy, repeated use of tanning beds, immunosuppressive therapy, acquired and congenital disorders of the immune system, and oncological diseases treated with aggressive chemotherapy or immunotherapy [28,29].

Clinical diagnosis is also based on the ABCDEF criteria, which in many cases will help to diagnose developing melanoma. These criteria include asymmetry of the lesion, borders irregularity, color variety, diameter larger than 5 mm, and firm (nodular) or funny-looking lesion or ugly duckling sign (different from the others) [26]. Cutaneous melanoma occurs as an irregular oval or polycyclic lesion with a typical variety of colors (brown, gray, blue-black, whitish, pink). In the beginning, horizontal growth of the tumor prevails, the progression will show vertical growth and the formation of the nodular lesion, with a tendency to ulcerate and bleed after minimal trauma which signals the growth of the tumor in depth and thus a worse prognosis of the disease. A rare amelanotic or a hypomelanotic subtype of melanoma due to lack of pigmentation can mimic other benign and malignant conditions [30,31]. Melanoma can occur anywhere on the skin, on the mucous membranes, in the iris of the eye, in the soft coverings of the brain, in the heart, in the urogenital tract, and in the lymph node [28].

Early surgical extirpation of the primary tumor with a sufficient safety margin with the extirpation of the adjacent subcutaneous tissue is currently the only curative treatment for this cancer. After a precise histological examination, the TNM staging and the further course of treatment are determined. If a complete examination (using sonography, CT, or PET/CT) does not detect metastatic spread of the disease, patients remain in follow-up with their oncologist/oncodermatologist. In case of evidence of metastases, adequate surgical treatment, radiotherapy, chemotherapy, or biological treatment is chosen. For unresectable or distant metastatic disease, a combination therapy with nivolumab/ipilimumab has been a recommended option for first-line, second-, or subsequent-line systemic therapy. Patients diagnosed with melanoma should be regularly examined [32,33,34].

## 3. Phenolic Compounds

Phenolic compounds are widely distributed throughout the plant kingdom and represent a significant class of secondary metabolites in plants. They can be found in various plant tissues, including fruits, nuts, grains, legumes, and spices. These compounds play essential roles in numerous physiological processes, such as improving plant quality, adding to the color and flavor profiles, and enhancing stress resistance. Phenolic compounds share a common chemical structure consisting of an aromatic ring accompanied by one or more hydroxyl substituents. These compounds can be further categorized into several classes, with the primary groups of phenolic compounds encompassing phenolic acids, flavonoids, tannins, stilbenes, coumarins, and lignans (Figure 1) [35].

*Phenolic acids*, which belong to the group of non-flavonoid phenolic compounds, are extensively found in plants in various forms, including free, conjugated-soluble, and insoluble-bound forms. They are commonly found in fruits, vegetables, whole grains, coffee, and spices. Phenolic acids in plants are associated with a wide range of functions, although much remains to be understood about their specific roles. These functions include facilitating nutrient uptake, promoting protein synthesis, enhancing enzyme activity, supporting photosynthesis, and contributing to structural components. Phenolic acids are derivatives of benzoic (e.g., 4-hydroxybenzoic acid, protocatechuic acid, vanillic acid, gallic acid, and syringic acid) and cinnamic (e.g., *p*-coumaric acid, caffeic acid, chlorogenic acid, ferulic acid, and sinapic acid) acids [36,37]. 

*Flavonoids* are a large class of phenolic compounds that include subclasses such as flavones, flavonols, flavanones, flavanols (catechins), anthocyanins, and isoflavones. They are widely distributed in fruits, vegetables, grains, and beverages like tea and wine. Flavonoids in plants serve multiple roles, including the regulation of cell growth, the attraction of pollinating insects, the protection of enzymes and vitamins, and providing protection against both biotic and abiotic stresses [38,39]. 

*Tannins* are a group of phenolic compounds that are commonly found in plant-based foods such as tea, coffee, cocoa, and fruits like grapes and pomegranates. They contribute to the astringent taste in these foods [40].

*Stilbenes* are a class of phenolic compounds that are characterized by the presence of a stilbene backbone. Several plants produce natural stilbenes as a defense mechanism against various pathological conditions, including excessive ultraviolet (UV) radiation, exposure to high temperatures, attacks by insects, as well as fungal or bacterial infections. Resveratrol is the most well-known stilbene and is found in grapes, berries, and red wine [41].

*Coumarins* are phenolic compounds with a benzopyrone structure. They can be found in various plants such as tonka beans, cinnamon, and sweet clover. They participate in the antimicrobial plant’s chemical defense strategy, in adaptive plant responses to iron deficiency, or in the interaction between plant roots and beneficial microbes [42].

*Lignans* are phenolic compounds found in plant sources such as flaxseeds, sesame seeds, whole grains, and berries. They are known for their potential health benefits, including antioxidant and hormone-balancing effects. Lignans in plants serve as defensive compounds, safeguarding them against potential threats from insects, microorganisms, and even neighboring plants [43].

Apart from their natural role, phenolic compounds exhibit notable pharmacological properties that encompass anti-inflammatory [44], immunosuppressive [45], cardioprotective [46], antioxidant [47], neuroprotective [48], and antimicrobial activities [49]. Moreover, several phenolic compounds have shown anticancer activity by inhibiting the growth of cancer cells [50], promoting cell death such as apoptosis [51], autophagy [52], necroptosis [53], or targeting numerous signaling pathways [54,55,56]. In addition, the formation of new blood vessels that support tumor growth has also been reported as a possible target of phenolic compounds [57,58,59].

## 4. Overview of Selected Spices

### 4.1. Allspice

Allspice (*Pimenta dioica*) is an evergreen tree belonging to the myrtle family (*Myrtaceae*). It is commonly referred to as Jamaican pepper or pimento. Originating from the West Indies and Central America, the allspice plant derives its name from the distinctive taste of its dried berries, which is reminiscent of a blend of cloves, cinnamon, and nutmeg. In folk medicine, Allspice berries have been used for the treatment of colds, dysmenorrhea, dyspepsia, diabetes, myalgia, and sore joints [60]. Furthermore, Allspice has been found to have different pharmacological activities such as antimicrobial, antioxidant, anti-inflammatory, or anticancer effects [61,62,63,64] due to the presence of numerous phytochemicals among them eugenol is a main bioactive constituent [65]. Other phenolic compounds such as methyl eugenol, isoeugenol, chavicol, flavonoids (e.g., quercetin, quercitrin, kaempferol, catechin, and naringenin), tannins (e.g., vascalaginone, grandininol), or phenolic acids (e.g., syringic acid, caffeic acid, coumaric acid, and cinnamic acid) have also been reported as active constituents of Allspice [66,67,68].

### 4.2. Alpinia galanga

*Alpinia galanga* (L.) Willd, also known as galangal, is a plant species in the ginger family (*Zingiberaceae*). It is native to Southeast Asia and is widely cultivated in Indonesia, Malaysia, Thailand, and other tropical countries [69]. Galangal rhizome is a widely utilized spice available in fresh, frozen, dried, or powdered forms [70]. In traditional medicine, it is used to treat a variety of ailments such as digestive problems, respiratory infections, rheumatic pain, fever, and skin diseases. It is also believed to have anti-inflammatory, antioxidant, and anticancer properties [71]. The analysis of the phytochemical constituents of *A. galanga* indicated the presence of numerous bioactive compounds such as alkaloids, tannins, terpenoids, saponins, and phenolics [72]. Regarding phenolic compounds, several flavonoids and non-flavonoid phenolic compounds are present in A. galanga. Flavonoids, such as galangin, galangin-3-methylether, kaempferol, quercetin, isorhamnetin, apigenin, kumatakenin, and pinocembrin have been isolated from the rhizomes and seeds of *A. galanga* [73]. In addition, numerous other, non-flavonoid phenolic compounds such as phenolic acids (e.g., ferulic acid, gallic acid), lignans (galanganal, galanganol A–C), and chalcones (galanganones A–C), and other phenolic compounds such as ellagic acid, 1′-acetoxychavicol acetate, methyleugenol, and many other have also been identified in *A. galanga* [74,75,76]. 

Numerous studies have documented the anticancer effect of compounds present in *A. galanga* [77,78,79]. Moreover, some of the A. galanga constituents also possess suppressive effects against skin cancer cells.

### 4.3. Black Cumin

*Nigella sativa* (L.), commonly known as black cumin or black seed, is an annual plant belonging to the *Ranunculaceae* family. It is cultivated for its highly aromatic seeds that are utilized as a spice and in traditional medicine. This plant, also referred to as black caraway, and is indigenous to South and Southwest Asia. Furthermore, it is prevalent in regions such as Northern Africa, the Middle East, and southern Europe [80]. The seeds and oil of this plant are widely used in traditional medicine for managing various ailments such as rheumatoid arthritis, asthma, inflammatory conditions, diabetes, and digestive disorders [81]. *Nigella sativa* is known to contain a range of active compounds, with thymoquinone being the most significant. Among others, several phenolic compounds have been isolated from black cumin seeds including phenolic acids (caftaric acid, *p*-hydroxybenzoic acid, syringic acid, protocatechuic acid, and chlorogenic acid) flavonoids (kaempferol, kaempferol-3-glucoside, quercetin, quercitrin, and diosmin), or other phenolic compounds (*p*-cymene, thymohydroquinone, thymol, carvacrol, and t-anethole) [81,82,83].

### 4.4. Black Pepper

Black pepper (*Piper nigrum*, L.) is a plant belonging to the *Piperaceae* family. The dried fruits of this plant are utilized as a spice across the globe. It originates from the Malabar Coast of India and it is one of the earliest known spices. It has a pungent, spicy taste and a distinct aroma, making it a popular seasoning and spice in cuisines all over the world. Black pepper is rich in bioactive compounds including alkaloids (piperine), sesquiterpenes (e.g., β-caryophyllene), and monoterpenes (e.g., limonene, β-pinene, α-pinene, sabinene, and camphene), oxygenated terpenes (e.g., linalool, terpinene-4-ol, and eugenol). Moreover, several phenolic acids (e.g., hydroxybenzoic acid, gallic acid, caffeic acid, and hydroxycinnamic acids) and flavonoids (e.g., quercetin, catechin, epicatechin, myricetin kaempferol, isoquercetin, and isorhamnetin) have been extracted from black pepper [84,85,86]. Black pepper is also used for medicinal purposes and has various health benefits including antioxidant, antimicrobial, anti-inflammatory, or gastroprotective effects [87]. Moreover, some bioactive compounds isolated from black pepper also have anticancer effects [88,89,90].

### 4.5. Cinnamon

Cinnamon, scientifically known as *Cinnamomum verum*, and commonly referred to as Ceylon cinnamon or true cinnamon, belongs to the *Lauraceae* family. This tree is native to Sri Lanka (formerly Ceylon), the neighboring Malabar Coast of India, and Myanmar. The spice is derived from the inner bark of the cinnamon tree. Other notable species within the same family include *C. cassia, C. burmannii, C. loureiroi,* and *C. citriodorum*. It is widely recognized as one of the most essential spices and is embraced and utilized by people across the globe in their daily lives [91]. Although cinnamon is preferably used as a spice, it also possesses several biological effects such as anti-tumor [92], anti-diabetic [93], anti-inflammatory [94], anti-microbial [95], and anti-oxidant activities [96]. The phytochemical content of cinnamon has shown the presence of numerous active constituents extracted from the leaf and bark oils with eugenol and cinnamaldehyde as the major constituents [97]. Among bioactive constituents, several phenolic compounds have been detected including the above-mentioned eugenol and cinnamaldehyde, phenolic acid (e.g., caffeic acid, chlorogenic acid, gallic, rosmarinic acid, *p*-coumaric acid, protocatechuic acid, *p*-hydroxybenzoic acid, and *trans*-vanillic acid) and flavonoids (e.g., rutin, apigenin, catechin, and epicatechin) [98,99]. Other phenolic compounds such as cinnamyl alcohol, α-terpineol, cinnamyl acetate, *p*-cymene, coumarin, and pyrogallol have also been found in cinnamon extracts [100].

The phenol commonly investigated in relation to skin cancers is eugenol, which is found abundantly in this spice.

### 4.6. Coriander

Coriander (*Coriandrum sativum* L), a plant from *Apiaceae* family, is one of the earliest spices used by humans. It grows wild over a wide area of Western Asia and southern Europe but today it is cultivated in The Netherlands, Central and Eastern Europe, China, India, Bangladesh, and North Africa [101]. In traditional medicines, it has been used for digestive and gastric troubles, fever, cough, dysentery, and many other complaints [102]. Analyses of the bioactive constituents of coriander showed numerous phytochemicals responsible for the therapeutic and nutritional effects of this plant including fatty acids, sterols, tocols, essential oils, and water-soluble compounds [103]. Moreover, many phenolic compounds have been isolated either from coriander seeds including phenolic acids (caffeic acid, chlorogenic acid, ferulic acid, gallic acid, o-coumaric acid, *trans*-hydroxycinnamic acid, *p*-coumaric acid, rosmarinic acid, salicylic acid, *trans*-cinnamic acid, and vanillic acid) and flavonoids (e.g., rutin, luteolin quercetin, kaempferol naringin, and apigenin) [104]. The primary active constituents discovered in the leaves of coriander were quercetin derivatives such as quercetin-3-O-rutinoside quercetin 3-O-glucuronide, quercetin-3-O-glucoside, and kaempferol-3-O-rutinoside [105]. In addition, numerous other phenolic compounds have been detected in the aerial parts of the coriander plant including esculin, esculetin, apigenin, luteolin, diosmin, catechin, orientin, vanillic acid, *p*-coumaric acid, *cis-* and *trans*-ferulic acid, gallic acid, maleic acid, and others [106,107]. However, it is important to mention that the constituents of coriander can differ depending on factors such as the variety, growth stage, planting location, and season of cultivation.

Several reports documented the antiproliferative effect of either coriander extract or its pure phytochemicals [108,109,110,111]. In addition, coriander extract significantly suppressed the migration and invasion of murine melanoma cells as well as the number of metastatic regions in mice bearing B16F10 melanoma cells [112]. Furthermore, some of the coriander’s bioactive components also showed the potential to modulate the activity of skin cancer cells.

### 4.7. Fenugreek

Fenugreek (*Trigonella foenum-graecum*) is a herbaceous plant that belongs to the *Leguminosae family*. It is commonly used as a spice and medicinal herb, and its seeds are utilized in various culinary preparations, herbal supplements, and traditional medicines [113,114,115]. Fenugreek is native to the Mediterranean region, southern Europe, and western Asia and is widely cultivated in many parts of the world for its culinary and medicinal properties [116].

In traditional medicine, fenugreek is used for its potential benefits in addressing various health conditions, including but not limited to cancer, hypercholesterolemia, diabetes, cold cough, splenomegaly, hepatitis, backache, and inflammation [117]. Phytochemical analyses showed that fenugreek contains a diverse array of secondary metabolites, which include saponins, steroids, alkaloids, flavonoids, terpenes, phenolic acid derivatives, amino acids, as well as fatty acids and their derivatives [118,119]. Related to phenolic compounds, several flavonoids (e.g., quercetin, luteolin, vitexin, isovitexin, kaempferol, tricin, and naringenin) and phenolic acids (e.g., *p*-coumaric acid, caffeic acid, and chlorogenic acid) have been isolated either from whole plants or seeds [120,121]. In the past decade, there has been extensive research on fenugreek and its active compounds as potential agents in the fight against cancer [122,123,124,125]. Regarding skin cancers, several phenolic fenugreek constituents have been reported to suppress the viability and growth of skin cancer cells either *in vitro* or *in vivo*. 

### 4.8. Ginger

Ginger is the rhizome of *Zingiber officinale* a plant belonging to the ginger family (*Zingiberaceae*). The same family also includes other plants used as spices, namely turmeric, cardamom, and galangal. It is native to southeastern Asia and was later introduced to other parts of the world. Ancient civilizations, including the Chinese, Indians, Greeks, and Romans, valued ginger for its medicinal benefits [126]. Ginger has a pungent, spicy flavor and a warm aroma, which is why it is a popular ingredient in various cuisines worldwide. In traditional medicine, ginger was commonly used to treat digestive disorders, including nausea, indigestion, and flatulence. It was also used as a warming herb to improve circulation and alleviate symptoms of colds, coughs, and respiratory ailments [127]. The chemical composition of ginger reveals the presence of more than 400 distinct compounds. Ginger is rich in terpenes such as cineole, citral, limonene, α/β pinenes, β-elemene, farnesene zerumbone, and zingiberene, and phenolic compounds including gingerols (e.g., 6-gingerol, 8-gingerol, and 10-gingerol) and shogalols (e.g., 6-shogaol, 8-shogaol, and 10-shogaol). In addition, several other phenolic compounds have been identified in ginger including quercetin, gingerenone-A, zingerone, 8-paradol, or 6-dehydrogingerdione [128,129,130]. Ginger extract or its constituents were reported to have a broad spectrum of pharmacological activities such as antioxidant, anti-inflammatory, antimicrobial, neuroprotective, cardioprotective, and gastroprotective properties [128,131]. Moreover, there are numerous studies focusing on the anticancer effects of ginger [132,133].

### 4.9. Oregano

*Origanum vulgare* L., also known as oregano, is a Mediterranean plant species belonging to the *Lamiaceae* family which, nowadays, represents one of the most used culinary herbs. However, the application of oregano in several ethnobotanical practices, including folk medicine, dates to ancient times. Regarding its phytotherapeutic effect, various investigations have been performed, documenting that oregano essential oil possesses antimicrobial, antiviral, antifungal, antioxidant, anti-inflammatory, digestive, expectorant, and neuroprotective effects. In addition, antiproliferative and anticancer effects of oregano have been reported [134,135]. Although the chemical composition of oregano strongly depends on the vegetative periods of the growing season, generally carvacrol and thymol are the two main components of this spice. In addition, terpenes (e.g., α-thujene, α-pinene, camphene, sabinene, β-pinene, β-myrcene, α-phellandrene, α-cubebene, β–elemene, *trans*-caryophyllene, eucalyptol, and linalool), flavonoids (e.g., rutin, naringin, hesperetin, naringenin, apigenin, luteolin, acacetin, and vitexin) or phenolic acids (caffeinic acid and rosmarinic acid) have also been identified in oregano extract or essential oils [136,137]. 

### 4.10. Nutmeg

Nutmeg is a spice derived from the seed of the *Myristica fragrans*, which is native to Indonesia. Now, for commercial purposes, it is widely cultivated in several countries, including India, Thailand, Japan, China, and South Africa. The spice is obtained by grinding the seed into a powder or grating it into a fine texture. Spice is commonly used in both sweet and savory dishes, as well as beverages. Additionally, nutmeg has been used in traditional medicine to treat anxiety, gastrointestinal discomfort, insomnia rheumatism, and also as an aphrodisiac [138,139]. Moreover, it has been used in several pharmacological actions such as antidiabetic [85], anti-inflammatory, analgetic [140,141], antimicrobial [142,143], hepatoprotective [144], or antioxidant [145] treatments. The effects of nutmeg are thought to be mediated by its bioactive constituents, which include lignans, neolignans, diphenyl alkanes, phenylpropanoids, terpenoids, alkanes, fatty acids, and fatty acid esters, as well as a few minor components like steroids, saponins, triterpenoids, and flavonoids. Phenolic compounds found in nutmeg encompass various categories such as phenolic acids (e.g., protocatechuic acid, caffeic acid, vanillic acid, *p*-coumaric acid, ferulic acid, and sinapic acid) and flavonoids (e.g., catechin, epicatechin, rutin, quercitrin, isoquercitrin, quercetin, and kaempferol), as well as other phenolic compounds like ellagic acid, myristicin, and elemicin [146,147]. 

### 4.11. Red Chili

Red chili (*Capsicum annuum* L.) known also as hot pepper, chili pepper, or red pepper is one of the most widely cultivated spices from the genus *Capsicum*. This plant probably originated from central-east Mexico [148]. Archaeological evidence suggests that chili peppers have been cultivated and consumed in this region for at least 6000 years. It has been utilized for centuries in different countries and civilizations. In ancient times, the Maya civilization employed pepper as a remedy for conditions such as asthma, coughs, and sore throats. Similarly, the Aztecs used it to alleviate toothaches. Chemical analyses of red chili showed a broad spectrum of active secondary metabolites. The spiciness of red chili peppers is primarily caused by capsaicinoids, which are alkaloids that are exclusively found in the *Capsicum* genus. Capsaicin and dihydrocapsaicin, among all capsaicinoids, account for approximately 77–98% of the pungency found in chili peppers. Other capsaicinoids include nordihydrocapsaicin, homodihydrocapsaicin, and homocapsaicin [149]. Furthermore, carotenoids responsible for the red color of chili peppers (capsanthin and capsorubin) or yellow-orange color (e.g., violaxanthin, zeaxanthin, β-cryptoxanthin, lutein, and β-carotene) have also been identified in a red chili fructi [150]. Among flavonoids, quercetin and luteolin have been described as the main flavonoids in *Capsicum* [151]. However, besides these two, several other flavonoids, including kaempferol, myricetin, apigenin, naringenin, catechin, epigallocatechin, and their derivatives, have also been identified in this spice [152]. Moreover, a number of phenolic acids such as gallic acid, protocatechuic acid, vanillic acid, hydroxyl cinnamic acids, caffeic acid, ferulic acid, chlorogenic acid, and cinnamic acid have been detected in red chili extract [153]. Capsaicin, a main active compound, exhibits a wide range of pharmaceutical applications due to its analgetic anti-arthritic and anti-inflammatory properties, as well as its effectiveness against bacterial infections, Moreover, it was also reported to suppress the growth of cancer cells [154,155]. 

### 4.12. Rosemary

*Salvia rosmarinus* Spenn. (syn. *Rosmarinus officinalis*), commonly known as rosemary, is a medicinal aromatic plant of the *Lamiaceae* family, native to the Mediterranean region. It is usually a small shrub with evergreen small needle-like leaves of light grey-green color. It is popularly used as a spice and flavoring. Although it is mostly preferred as a spice in food, this herb is one of the most popular medicinal herbs in the world. Rosemary is used in folk medicine for wound healing, in treating inflammatory diseases and mycoses, and to relieve renal colic, muscle spasms, dysmenorrhea, and headaches. It also has antiviral, antibacterial, antithrombotic, antidepressant, and antioxidant effects [156,157]. Chemical analyses showed a broad spectrum of bioactive constituents. Numerous compounds were identified in the essential oil of rosemary. Among them, the predominant constituents were 1,8-cineol, camphor, α-pinene, limonene, camphene, and linalool, respectively [158]. Furthermore, several phenolic compounds have been identified in rosemary extract including phenolic acids (salvianic acid, caffeic acid, rosmarinic acid, and salvianic acid A), flavonoids (luteolin, luteolin−7-O-rutinoxide, luteolin-7-glucoronide, hesperidin, apigenin, cirsimaritin, genkwanin, and salvigenin), and phenolic diterpenes including rosmadial, 7-methylrosmanol, carnosol, carnosic acid, and 12-methylcarnosic acid [159]. In addition to above-mentioned biological effects, several studies have specifically investigated the mechanisms underlying the anticancer effects of rosemary or its active constituents [160,161].

### 4.13. Saffron

*Crocus sativus*, commonly known as saffron, is a flowering plant in the *Iridaceae* family. It is cultivated for its highly prized saffron spice, which is derived from the dried stigmas of the flowers. Saffron is known for its vibrant red color, distinct aroma, and unique flavor. This plant is primarily cultivated in regions with a Mediterranean climate, such as Iran, Spain, India, and Greece [162]. Apart from its culinary uses, saffron has also been traditionally valued for its medicinal properties. In traditional medicine, it has been used for antioxidant, anti-inflammatory, anti-vomiting, and mood-enhancing effects [163]. Analyses of its chemical composition showed that saffron contains numerous active constituents belonging to carotenoids such as crocin, crocetin, and lycopene, monoterpenes including picrocrocin, crocusatins, isophorone, and amino acids, and alkaloids (e.g., 5-methyluracil, pyridin-3-ylmethanol, uracil, harman, and tribulusterine). Moreover, saffron also contains numerous phenolic compounds including phenolic acids (e.g., protocatechuic acid, 4-hydroxybenzoic acid, vanillic acid benzoic acid, and *p*-coumaric acid) or flavonoids and flavonoid glycosides (e.g., kaempferol, kaempferide, kaempferol-3-O-sophoroside-7-O-glucoside, kaempferol-3,7,4′-triglucoside, kaempferol 7-O-β-D-glucopyranoside, isorhamnetin-3,4′-diglucoside, isorhamnetin-3-O-glucoside, astragalin, sophoraflavonoloside, and helichrysoside) [164].

### 4.14. Sichuan Pepper 

Plants belonging to the genus *Zanthoxylum* L. (the fruits of this genus are known as Sichuan pepper) form the largest group within the *Rutaceae* family and encompass approximately 250 species, including *Z. bungeanum, Z. zanthoxyloides, Z. ovalifolium, Z. armatum*, and many others [165]. These are deciduous, heat-loving, evergreen plants found in subtropical and warm climates worldwide, particularly in Asia, Africa, and America. Traditionally, they have been used as food and for their positive effects on the human body in the prevention and treatment of various diseases in countries like China, Japan, India, and Vietnam [166]. As a spice, not only the fruits but also the seeds, leaves, roots, and bark are utilized. It has a pungent aroma, and a citrus–floral taste, and causes a tingling sensation in the mouth. Despite being commonly referred to as “pepper”, it is not related to peppercorn from the *Piperaceae* family [167]. Current research pays significant attention to this genus. Many of its effects have been demonstrated, including anti-tumor, neuroprotective, antiparasitic, antimicrobial, antiviral, antioxidant, anti-inflammatory, antidiabetic, hepatoprotective, anti-obesity, and other properties [166,167,168]. These effects are mediated by bioactive compounds, among which volatile odor-bearing alcohols, ethers, terpenoids, and related compounds, as well as non-volatile alkyl amides, polyphenols, and their glycosides. Polyphenols include isovitexin, vitexin, hyperoside, isoquercitrin, rutin, foeniculin, trifolin, quercitrin, astragalin, and afzelin [169].

### 4.15. Star Anise

*Illicium verum* Hook f., also known as badian, is an aromatic evergreen tree belonging to the *Illiciaceae* family [170]. It is named for its typically star-shaped fruit. It is naturally found almost exclusively in southern China and Vietnam but is also commercially grown in other Asian and European countries [171]. Star anise has a long history in China as both a spice and medicinal plant, listed in the Compendium of Materia Medica (Bencaogangmu) during the Ming Dynasty and in the Chinese Pharmacopoeia [170,172]. It is used in various forms, such as raw, dried, powdered, or as an essential oil. Traditionally, it has been used as a carminative, digestive, and spasmolytic for abdominal pain, colic, and vomiting, as a sedative for nervousness and insomnia, as an analgesic and anti-inflammatory for lower back pain, joint pain, and rheumatism, and also for symptoms of the common cold [170]. Its pleiotropic effects, including antimicrobial, antifungal, antiviral, antioxidant, anti-inflammatory, gastroprotective, and others are due to phytochemicals naturally occurring in star anise fruits [173,174,175,176]. Chemical analyses have shown that fruits contain significant amounts of terpenes, alkaloids, essential oils, and tannins. The most abundant phenolic compounds are phenylpropanoids *cis*- and *trans*-anethole (85–95%), estragole, anisylacetone, ρ-anisaldehyde, foeniculin, and others [177]. Additionally, it is a valuable source of shikimic acid which is an essential intermediate for the synthesis of the antiviral drug oseltamivir (Tamiflu^®^) [172]. In addition to the aforementioned health benefits, an anticancer effect of *Illicium* verum has been demonstrated in various *in vitro* and *in vivo* tumor models [178,179,180,181,182,183].

### 4.16. Sumac

Sumac, scientifically referred to as *Rhus coriaria* L., is a plant belonging to the *Anacardiaceae* family and is native to southern Europe. Its dried fruits are widely employed as a spice. Traditionally, sumac has been commonly utilized for addressing conditions such as gout, obesity, bleeding disorders, and diabetes. Its rich history of traditional usage aligns with its medicinal applications, making it a popular choice for these ailments [184]. Moreover, *Rhus coriaria*’s phytochemicals have been reported to have several pharmacological effects including antioxidant, anti-inflammatory, antimicrobial, antiemetic, hypolipidemic, or analgesic properties [185,186]. In addition, the anticancer effect of sumac against various types of cancers has also been reported [187,188,189]. The chemical analyses showed that sumac contains numerous active compounds including essential oils, steroids, terpenoids, and organic acids. However, the dominant groups of bioactive compounds found in sumac are phenolic compounds, mainly ydrolysable tanins (e.g., galloylhexose, O-galloylmorbergenin, and O-galloyl arbutin), flavonoids (e.g., quercetin, myrecetin, apigenin, kaempferol, and quercetin 3-glucoside), phenolic acids (gallic acid, ellagic acid, protocatechuic acid), and anthocyanins (e.g., delphidin-3-glucoside, cyanidin 3-(2″-galloyl)galactoside, and cyanidin-3-glucoside) [184,190,191,192].

### 4.17. Thyme

*Thymus vulgaris* L. known as thyme (the Greek word ‘thymos’ means strength or courage) belongs to the *Lamiaceae* family. This plant grows in dry and hot climates in southern Europe, North Africa, and several parts of Asia. The genus Thymus includes approximately 300 species distributed worldwide. Thyme has been widely used as a spice. Its essential oils are used in traditional medicine for their antiseptic, anthelmintic, expectorant, antispasmodic, antimicrobial, antiviral, antifungal, antioxidant, sedative, as well as antirheumatic, antihypertensive, and calming effects. Therefore, it is used to treat a variety of ailments (respiratory diseases, prevention of atherosclerosis, urinary tract infections, dyspepsia, toothache, and many others). Its aromatic extracts are also used in cosmetics [193,194]. In addition, several authors reported the potential of either thyme extract or thyme bioactive compounds to suppress the growth and viability of different cancer cells [195,196,197]. Analyses of chemical composition revealed that thyme contains abundant secondary metabolites of which thymol and carvacrol are the major constituents. Others include terpenoids (e.g., β-, α-fellandrene, *p*-cymene, m-cymene, eucalyptol, and *trans*-caryophyllene), phenolic acids (e.g., caffeic acid, quinic acid, *p*-coumaric acid, ferulic acid, cinnamic acid, and rosmarinic acid), and flavonoids (e.g., apigenin, naringnin, and luteolin-7-o-rutinose) [198,199]. 

### 4.18. Turmeric

The ginger family (*Zingiberaceae*) includes the perennial herbaceous plant known as turmeric (*Curcuma longa*). Since antiquity, several uses have been made for its underground stems, known as tuberous rhizomes. They are used as a common condiment, a natural textile dye, and have therapeutic benefits as an aromatic stimulant. In addition to being a spice, turmeric has also been used historically as a perfume. It comes from southern India and Indonesia. Due to its flavor and color, turmeric is a common spice used in the creation of curries in India and other Asian nations [200]. Turmeric has been used for treating coughs, diabetic wounds, hepatic diseases, cardiovascular illness, dyspepsia, and epigastric discomfort for a long time in Chinese and Ayurvedic medicine. A study focusing on turmeric composition led to the extraction of more than one hundred and fifty secondary metabolites. These compounds comprised a diverse range of chemical classes including monoterpenoids, sesquiterpenoids, diterpenoids, and triterpenoids. Furthermore, the phytochemical analysis of turmeric also revealed the presence of saccharides, steroids, fatty acids, and alkaloids among its constituents. Moreover, turmeric is a rich source of phenolic compounds including curcuminoids (e.g., curcumin, demethoxycurcumin, and bisdemethoxycurcumin), phenolic acids (e.g., gallic acid, protocatechuic acid, chlorogenic acid, cinnamic acid, and ferulic acid), flavonoids (e.g., rutin, myricetin, genistein, quercetin, catechin, and epicatechin) and other phenylpropanoids (e.g., coumarin) [201,202]. 

In addition to its various biological effects, the potential of turmeric and its active constituents as anticancer agents has been explored in recent research studies [203,204]. Furthermore, several of the aforementioned phenolic compounds have been investigated for their potential as agents against skin cancers [205,206,207,208].

Table 1 provides the summary of phenolic compounds identified in the above-mentioned spices.

## 5. Effect of Spices-Derived Phenolic Compounds against Melanoma and Non-Melanoma Skin Cancer 

### 5.1. Phenolic Acids

The chemical structures of phenolic acids discussed in this review are shown in Figure 2.

**Caffeic acid** (CA) is widely distributed in nature and can be found in various plants, including coffee beans, fruits, vegetables, herbs, and spices. Caffeic acid has been studied for its potential in the prevention and treatment of skin cancer due to its antioxidant and anti-inflammatory properties. In the study conducted by Yang et al. (2014), CA in human A431 skin cancer cells suppressed colony formation and in non-cancer HaCaT cells prevented EGF-induced neoplastic transformation. Moreover, in animal experiments, it significantly decreased tumor incidence and volume in a solar UV (SUV)-induced skin carcinogenesis [209]. Furthermore, Western blot analyses demonstrated that CA effectively suppressed ERK1/2 activities both *in vitro* and *in vivo*. This indicates that CA may exert chemopreventive effects against SUV-induced skin carcinogenesis by specifically targeting ERK1/2 signaling pathways. Furthermore, previous research demonstrated that the protective effect of CA against UV-induced skin damage is also linked to its ability to counteract ultraviolet B (UVB)-triggered immune suppression. Topical application of CA to the skin of mice exposed to UVB radiation, significantly reduced the UVB-induced increase of IL-10 mRNA expression as well as JNK and p38 kinases activation, factors contributing to the development of skin cancer [210]. Later, the studies conducted by Agilan et al. [211,212] showed that several other mechanisms can be involved in the skin-preventive effect of CA. It was found that in the UVB-irradiated animals, CA significantly suppressed the UVB-induced activation of the JAK-STAT3 axis followed by the activation of the apoptotic machinery in mouse skin. Moreover, CA prevented a decrease of UVB-induced loss of TSP-1, a naturally occurring antiangiogenic protein inhibited the proliferative markers such as PCNA, cyclin-D1, or Bcl-2. Furthermore, caffeic acid exhibited a reduction in UVB-induced skin inflammation, as evidenced by lowered lipid peroxidation and a reduced expression of inflammatory mediators like TNF-α, IL-6, and COX-2. This effect was accompanied by the activation of peroxisome proliferator-activated receptors within the mice skin. Additionally, whether applied topically or systemically, caffeic acid led to a decrease in the occurrence and number of SCC in experimental animals subjected to 30 weeks of UVB exposure.

In JB6 P+ mouse skin epidermal cells, a separate study revealed that CA demonstrated the capability to inhibit UVB-induced COX-2 expression. This inhibition was achieved by suppressing the activity of Fyn kinase, which is a crucial player in skin carcinogenesis. Moreover, caffeic acid also blocked Fyn kinase downstream MAPKs contributing to its potential as a preventive agent against skin carcinogenesis caused by UVB exposure [213]. Additionally, it has been shown that CA effectively inhibits UVB-induced DNA damage, reduces the generation of ROS, and decreases the frequency of apoptotic cell death in human dermal fibroblasts. Moreover, it has been found to prevent the UVB-induced expression of PI3K and AKT kinases by activating PTEN, a tumor suppressor protein. This suggests that CA may possess protective properties against UVB-induced skin damage and potentially contributes to skin health and anticancer effects through these mechanisms [214]. Other authors documented the antiproliferative effect of CA in melanoma cells. Decreased cell viability, increased number of dead cells, and inhibited colony formation have been found in CA-treated human SK-Mel-28 melanoma cells. In addition, it increased the percentage of apoptotic cells and increased the number of cells at the G0–G1 phase of the cell cycle [215].

In another study, CA and **chlorogenic acid** (ChA) induced apoptosis in amelanotic melanoma C32 cells. This effect was potentiated with static magnetic field (SMF) exposition. Caffeic acid also significantly upregulated mRNA expression of the Casp3 and caspase-3 activity in C32 cells [216]. Later, they reported that both phenolic acids in the same melanoma cell line in combination with SMF decreased TGFβ protein levels [217]. Moreover, ChA in C32 cells suppressed the viability of melanoma cells and the expression of genes of antioxidant enzymes superoxide dismutase (SOD1, SOD2) and glutathione peroxidase (GPX1), as well as activity of SOD, GPx, and catalase (CAT). Surprisingly, in combination with SMF, the activity of antioxidant enzymes was increased [218].

***p*-Coumaric acid** (*p*-CA), another member of the phenolic acids, has a broad spectrum of biological activities including anti-inflammatory, antioxidant, antidiabetic, and hepatoprotective properties [219,220,221,222]. Moreover, the anticancer effect of *p*-CA has also been reported [223,224]. Regarding skin cancer, this phenolic acid decreased both human and murine melanoma cell viability and proliferation. The antiproliferative effect of *p*-CA has been associated with the cell-dependent arrest of the cell cycle—at the S phase in human A375 cells and at the G0–G1 phase in murine B16 cells. In human melanoma cells downregulation of cyclin A and CDK2 while in murine cells cyclin E and CDK2 have been observed. In addition, *p*-CA induced apoptosis in both melanoma cell lines [225]. In a recent study conducted by Gastaldello et al. (2021), gavage administration of *p*-CA to Balb/C mice bearing B16F10 melanoma cells resulted in a significant decrease in tumor volume and a reduction in the number of mitoses in the tumor region [226]. Moreover, *p*-CA also modulated some factors in the tumor microenvironment such as angiogenesis and inflammation. 

**Ferulic acid** (FA) has been evaluated as a perspective compound for skin cancer prevention. Ultraviolet B radiation can cause chronic skin inflammation which can result in the development of non-melanoma skin cancer, particularly basal cell carcinoma (BCC) and squamous cell carcinoma (SCC) [227]. In the context of UVB-induced skin inflammation, reducing inflammation can help minimize the risk of cancer formation by mitigating the inflammatory response and subsequent DNA mutations. In the study by Stanifort and co-workers (2012), topically applied FA attenuated UVB-induced changes in the skin of experimental animals [228]. Simultaneously, FA caused the degradation of matrix metalloproteinase-2 and -9 (MMP-2 and MMP-9) overexpressed after UVB irradiation. Later, it was documented that the protective effect of FA in UVB skin carcinogenesis was associated with the modulation of angiogenesis, inflammation, and apoptosis. UVB irradiation increased the expression of several angiogenic and inflammatory markers such as iNOS, IL-6, TNF-α, or VEGF. Moreover, the expression of anti-apoptotic Bcl-2 protein was upregulated while anti-apoptotic Bax was downregulated. The administration of FA before the UVB exposition significantly reverted all of these events [229]. Another study showed that FA effectively also prevented dimethylbenz[a]anthracene (DMBA)-induced carcinogenesis in experimental mice. Oral, but not topical, administration of FA prevented the formation of skin cancers, decreased lipid peroxidation and prevented decrease of antioxidants in the skin of animals [230]. 

**Gallic acid** (GA), a phenolic compound broadly distributed in the plant kingdom, has been reported to possess numerous biological activities [231]. Regarding skin cancers, GA has been reported to induce apoptosis in A375.S2 human melanoma cells. Molecular analyses showed that GA induced both caspase-dependent and caspase-independent pathways of apoptosis [232]. Subsequently, these researchers found that GA can hinder the invasiveness of melanoma cells by inhibiting MMP-2 through its involvement in the Ras and ERK signaling pathways [233]. A great proteomic analysis performed in the study by Liu et al. (2014) revealed the co-occurrence of apoptosis and modulation of glycolysis in the GA-treated B16F10 melanoma cells [234]. It was shown that gallic acid stimulated the mitochondrial apoptosis pathway and simultaneously upregulated several proteins involved in glycolysis including glucokinase, pyruvate kinase, glyceraldehyde-3-phosphate dehydrogenase α-enolase, and aldolase. In vivo findings also provided evidence that the topical application of GA yielded notable preventive effects against cancer development within a two-step skin carcinogenesis model involving DMBA and croton oil. GA co-treatment resulted in significant reductions in tumor incidence, the number of tumors per animal, and tumor volume. Additionally, GA effectively counteracted the severe hyperplasia triggered by DMBA/croton oil treatment. Furthermore, GA played a role in restoring the serum levels of LDH, GSH, and GSH, along with skin levels of antioxidants such as SOD, catalase, and GPx, to their normalized states. Notably, GA exhibited a substantial reduction in the activity of MMP-2 and MMP-9 induced by DMBA/croton oil treatment [235].

**Protocatechuic acid** (PA), a natural phenolic compound, is commonly found in various plant-based foods, such as fruits, vegetables, certain medicinal plants, and spices. In the study by Tseng and co-workers (1998), PA significantly inhibited the incidence of chemically induced skin tumors and simultaneously inhibited the activity of epidermal ornithine decarboxylase and myeloperoxidase as well as the formation of hydrogen peroxide in the mouse skin [236]. On the other hand, Nakamura et al. (2000) found that PA at a low dose (16 nM) decreased the number of tumors, and, surprisingly, at a higher dose (1600 nM) it increased the number of tumors [237]. The authors later verified that the effect of PA was dose-dependent. The application of this phenolic acid at a dose of 16 nM significantly prevented inflammation and oxidative stress. However, at a higher dose of 1600 nM, it had the opposite effect and actually intensified both inflammation and oxidative stress [238]. In addition, a study conducted by Lin et al. (2010) revealed the anti-metastatic potential of PA [239]. It significantly decreased the number of liver metastasis in mice injected with B16-F10 melanoma cells. Furthermore, levels of MMP-2, as well as the expression of Ras and phosphorylated Akt, were downregulated in PA-treated animals.

**Rosmarinic acid** (RA) is another phenolic acid found in several spices, which has decreased viability and suppressed migration of metastatic SK-MEL-28 melanoma cells. Rosmarinic acid-induced apoptosis was linked to increased caspase-8 and caspase-3 gene expression together with increased activity of caspase-3. Moreover, decreased gene expression of NLRP3 inflammasome has also been observed [240]. In another study, RA showed an inhibitory effect on melanoma cells via suppression of the ADAM17/EGFR/AKT/GSK3β axis resulting in the inhibition of melanoma cell proliferation, invasion, and migration. In addition, RA also enhanced the sensitivity of melanoma cells to cisplatin [241]. Additionally, the radiosensitizing effect of RA in melanoma B16F10 cells was documented and this effect can be associated with decreased GSH-dependent protection against oxidative stress [242,243]. Moreover, oral administration of RA completely prevented DMBA-induced carcinogenesis in Swiss albino mice. Histopathological evaluation showed mild dysplasia and hyperplasia but no tumor formation in DMBA and RA-treated animals. In addition, RA effectively counteracted the DMBA-induced excessive production of TBARs and mitigated the decline in antioxidant levels, encompassing SOD, CAT, GPx, and GSH, within the skin tissue of the experimental animals [244].

**Syringic acid** (SA), a phenolic compound found in numerous plants, has been reported to exhibit a wide range of biological activities such as antioxidant, anti-inflammatory, antimicrobial, and anticancer properties [245,246,247]. Regarding skin cancers, SA remarkably inhibited UVB-induced tumor incidence. Molecular analyses in human epidermal keratinocytes exposed to UVB irradiation showed that SA significantly downregulated the expression of COX-2, MMP-1, and prostaglandin E2. Additionally, this phenolic acid inhibited MAPK and EGFR phosphorylation induced by UVB [248].

Molecular and cellular mechanisms of the antiproliferative action of phenolic acids are listed in Table 2.

### 5.2. Flavonoids

Figure 3 presents chemical structures of flavonoids discussed in this review.

**Apigenin**, similar to many other flavonoids, has been found in different fruits and vegetables and a number of studies documented its broad spectrum of biological activities [249] its including anticancer effects [250]. In addition, a number of studies were focused on its anticancer effect against both melanoma and non-melanoma skin cancers. In the study conducted by Zhao and co-workers (2017), apigenin decreased viability, migration, and invasion in A375 and C8161 human melanoma cells [251]. Moreover, cell cycle arrest at the G2/M phase and apoptosis induction has been promoted in apigenin-treated cells. In addition, apigenin decreased the activation of several proteins including ERK 1/2, AKT, and mTOR. Additionally, the apigenin-induced anti-metastatic effect was associated with the modulation of STAT3 activity. In murine melanoma B16F10 cells, apigenin decreased STAT3 phosphorylation and STAT3 nuclear localization resulting in the suppression of STAT3 transcriptional activity. Furthermore, apigenin was observed to decrease the expression of genes targeted by STAT3, including MMP-2, MMP-9, VEGF, and Twist1, which play a role in cell invasion and migration [252]. In another study, it was shown that apigenin induced apoptosis in the A375SM human melanoma cells via upregulation of phosphorylated p38 with simultaneous downregulation of ERK, JNK, and Akt, as well as the kinases involved in the proliferation and growth of melanoma cells [253]. Additionally, it was reported that the antiproliferative and anticancer effects of apigenin can be associated with immune response modulation. Xu et al. (2018) found that apigenin suppressed PD-L1 expression in melanoma cells that have become more sensitive to cytotoxic T cells [254]. In addition, the same effect was observed in apigenin-treated host dendritic cells resulting in an augmented host immune response.

In experimental animals, apigenin has been found to prevent UVB-induced skin inflammation and carcinogenesis via different mechanisms. In the study by Mirzoeva and co-workers, apigenin suppressed carcinogenesis in mice exposed to UVB irradiation due to maintenance of the expression of thrombospondin-1, which serves as a natural inhibitor of angiogenesis and tumorigenesis [255]. Furthermore, other targets such as mTOR, Src kinase, and COX-2 are also suggested [256,257,258]. In addition, in the study conducted by Caltagirone et al. (2000), apigenin was found to delay tumor growth in mice bearing B16-BL6 melanoma cell xenograft [259]. Moreover, apigenin also significantly decreased the number of colonies in the lung which suggest the anti-invasive and anti-metastatic potential of this flavonoid. Later, in the study conducted by Kiraly and colleagues, it was observed that the topical application of apigenin at a concentration of 20 µmols led to a noteworthy decrease in the occurrence of skin cancer induced by DMBA/TPA, with rates dropping from 70% to 21%. Moreover, apigenin postponed the onset of tumor appearance as well as reduced tumor multiplicity. Furthermore, apigenin exhibited a significant inhibitory effect on COX-2 expression and activity in the skin of DMBA/TPA-treated animals, resulting in decreased levels of PGF2 and reduced expression of prostaglandin receptors EP1 and EP2 [260].

**Astragalin** (kaempferol-3-glucoside) is a significant component of plant extracts that have demonstrated antioxidant, anti-inflammatory, antimicrobial, antidiabetic, antidepressant, and anticancer effects [261]. The cytotoxic effect of astragalin on A375P and SK-MEL-2 melanoma cells was evidenced by an increase in the sub-G1 population and TUNEL-positive cell population, indicating the presence of DNA fragmentation in astragalin-treated cells. Cell death was accompanied by a decrease in cyclin D1 expression and the anti-apoptotic Bcl-2 protein Mcl-1, upregulation of Bax, and activation of caspase-3. A specific pro-apoptotic effect of astragalin includes a significant reduction in SOX10 expression in non-transfected cells as well as in cells overexpressing SOX10 [262]. SOX10 (Sry-related HMg-Box Gene 10) is a transcription factor that facilitates neural crest cell development and is also involved in melanoma cell growth through the regulation of immune checkpoint protein expression [263].

**Catechin** and its related molecules, such as epigallocatechin-3-gallate (EGCG) and epicatechin-3-gallate (ECG) have been found to possess antiproliferative and anticancer properties, including in skin tumors [264,265,266]. The main mechanism of their action is the induction of apoptosis and cell cycle arrest. For example, in A375 cells, in addition to mitochondrial damage, changes in the level of the Bcl-2 protein, and activation of caspase -3, catechin-induced modulation of autophagy has also been observed. Increased PI3K, Akt, and mTOR phosphorylation activated this signaling pathway that suppresses autophagy. Accordingly, reduced levels of other autophagic proteins such as Beclin-1, LC3, and phospho-AMPK were observed [267]. Immunotherapy is also a significant treatment approach for melanomas, and research on natural substances with anticancer effects in this area is continuously advancing. A study by Ravindran Menon et al. (2021) investigated the influence of EGCG on immune responses and the tumor microenvironment in human metastatic melanoma cell lines 1205Lu, HS294T, and A375 [268]. It was found that EGCG inhibited IFN-γ-induced PD-L1/PD-L2 expression. This effect depended on the suppression of JAK/STAT signaling, which mediates the interaction between cytokine receptors and the cell nucleus. Similar to *in vitro* experiments, *in vivo* inhibition of the JAK/STAT pathway led to reduced tumor growth in mice, decreased Ki-67-positive cells, and increased expression of granzyme in CD8+ cells in the tumor microenvironment. CD8+ T cells were found to be essential for suppressing tumor growth, and treatment with EGCG reactivated the immune response against the tumor even more effectively than anti-PD-1 antibody treatment.

**Chrysin** has also been reported to suppress melanoma cell proliferation and invasion. In human A375.S2 melanoma cells, chrysin inhibited cell migration and invasion via matrix metalloproteinase-2 (MMP-2) inhibition, and due to the inhibition of expression or activation of several signaling pathways including phosphoinositide 3-kinase (PI3K), protein kinase B (AKT), protein kinase C (PKC), focal adhesion kinase (FAK), Ras homolog family member A (RhoA), and *p*-c-Jun. Moreover, the downregulation of N-cadherin and upregulation of E-cadherin indicated that chrysin can also affect epithelial–mesenchymal transition [269]. Another study documented the antiproliferative and anticancer effects of chrysin *in vitro* and *in vivo* [270]. In murine B16F10 melanoma cells, exposition to chrysin led to the induction of cell cycle arrest at the G2/M phase associated with apoptosis induction. Moreover, in BALB/c mice bearing melanoma, xenograft chrysin decreased the size and weight of tumors after 21 days of treatment. In addition, the cytotoxic activity of NK cells, cytotoxic T lymphocytes, and macrophages has been found in chrysin-treated animals. Results of another study showed that chrysin induced apoptosis in human and murine melanoma cells through caspase activation and modulation of the activity of MAPK members including extracellular signal-regulated kinase (ERK) 1/2 and p38 MAPK [271].

**Diosmin**, a naturally occurring flavonoid identified in some spices, has been found to act synergistically with IFN-α in the treatment of metastatic pulmonary melanoma [272]. Diosmin is also able to induce apoptosis in non-melanoma skin cancer A431 cells associated with increased ROS generation, downregulation of Bcl-2, MMP-2 and MMP-9, as well as upregulation of p53, caspase-3, and caspase-9 [273].

In the above-mentioned study [274], diosmin significantly reduced the number of lung metastasis as well as growth and invasion index in animals bearing B16F10 cell xenograft. In another study, these authors compared the ability of diosmin, grape seed extract, and red wine to suppress the growth of lung metastasis in B16F10 melanoma cells inoculated in experimental animals. Among studied agents, diosmin showed the greatest reduction in pulmonary metastases [275]. 

**Galangin** has been documented to suppress the proliferation of the murine melanoma B16F10 cells in a dose-dependent manner. Moreover, galangin significantly reduced the ability of melanoma cells to form colonies indicating its antimetastatic potential as well. This effect has been associated with decreased cell adhesion to fibronectin and decreased cell motility and migration. In addition, galangin inhibited lung metastasis in mice bearing B16F10 xenografts [276]. Later, Benguedouar and co-workers (2016) presented that galangin-induced suppression of melanoma cell growth was associated with the induction of autophagy and apoptosis [277]. In addition, it was shown that galangin rescued human keratinocytes from undergoing apoptosis triggered by UVB-induced oxidative stress by restoring mitochondrial function and reducing the levels of apoptotic proteins [278]. In another study, they found that galangin in human keratinocytes increased the expression of GSH-synthesizing enzymes via activation of ERK/AKT-Nrf2 (nuclear factor erythroid 2-related factor 2) signaling [279]. These results indicate the possible preventive effect of galangin against UVB-induced skin carcinogenesis.

**Genistein** (4′, 5, 7-trihydroxy isoflavone) is a natural isoflavone and phytoestrogen mostly found in soybeans and also in other foods—alfalfa and clover sprouts, barley meal, broccoli, cauliflower, and sunflower, caraway, and clover seeds [280].

It has a variety of pharmacological activities including anti-inflammatory, antioxidant, and anticancer properties. Furthermore, the anticancer effect of this isoflavone has also been intensively studied [281,282]. Regarding melanoma cells, genistein was referred to inhibit human and animal melanoma cell growth and proliferation. These effects were associated with cell cycle arrest at the G2/M phase and cell differentiation promotion [283]. As reported by Darbon and co-workers (2000), G2/M arrest in human melanoma cells was associated with impaired dephosphorylation of Cdk1 and checkpoint kinase Chk2 activation [284]. Through detailed molecular analyses, the actions of genistein have been found to operate through a multifactorial mechanism. It is known that activation of the FAK/paxillin pathway and mitogen-activated protein kinases (MAPKs) signaling may serve as a possible indicator of melanoma metastasis. Genistein in B16F10 cells inhibited proliferation, invasion, and cell migration, and induced apoptosis in a dose-dependent manner. Moreover, genistein significantly suppressed the expression of numerous proteins including a phosphorylated form of FAK, paxillin, p38, ERK, and JNK, as well as the expression of tensin-2, vinculin, and α-actinin. Additionally, real-time PCR showed decreased gene expression of FAK and paxillin. Interestingly, it has been observed that low concentrations of genistein can activate the FAK/paxillin and MAPK signaling pathways, leading to enhanced invasion and migration of melanoma cells [285]. Another study reported the ability of genistein to inhibit the growth of A375SM melanoma cells via modulation of p21, cyclin E, and/or cyclin B genes expression and stimulation of ROS production followed by activation of p53 and p38 MAPK and activation of the ER stress-mediated apoptotic pathway [286]. Another author showed that genistein inhibited basal and PGE2-induced proliferation of melanoma cells and this effect was associated with the suppression of IL-8 and EP3 receptor expression [207]. Moreover, genistein has been shown to increase the sensitivity of melanoma cells to cisplatin, as their combination has significantly induced apoptosis while suppressing the expression of antiapoptotic proteins [287]. In addition to melanoma cells, genistein has also been reported to restrict cell growth in the HN4 squamous cell carcinoma due to blocking of the cell cycle at the S/G2-M phases and apoptosis induction in a time- and dose-dependent manner. Molecular analyses showed the downregulation of Cdc25C which plays an important role in the activation of the cyclinB1-Cdk1 complex [288].

In addition to *in vitro* studies, several *in vivo* experiments confirmed the anticancer effect of genistein in the xenograft model of carcinogenesis, chemically induced skin carcinogenesis, or UV-induced skin cancer. Genistein significantly prevented the growth of tumors in the female C57BL6J mice subcutaneously injected with B16 melanoma cells. Tumor volume was significantly reduced (by 50%) in animals feeding on a diet with genistein compared to the untreated group [289]. Furthermore, genistein suppressed the growth of lung metastasis in C57BL/6 mice-inoculated B16F-10 melanoma cells. Moreover, genistein significantly prolongs the survival of tumor-bearing animals [290]. In addition, in the study conducted by Farina and co-workers (2006), it was found that genistein in experimental animals injected with B16F0 melanoma cells significantly inhibited the formation of new blood vessels by tumor implants [291]. In another study, Wei et al. reported that genistein significantly decreased the incidence and multiplicity of skin tumors in two-stages of tumor models when DMBA was used as an initiator and 12-O-tetradecanoyl phorbol-13-acetate (TPA) was used as a promotor of skin carcinogenesis [292]. Genistein significantly reduced the DMBA-induced formation of DNA adduct as well as reduced TPA-induced H2O2 formation and inflammatory reaction. Furthermore, the preventive effect of genistein on UV-induced skin carcinogenesis has also been documented. Later, they reported a protective effect of genistein in UVB-induced skin carcinogenesis in mice. Genistein was applied either orally or topically. They found that genistein inhibited tumor incidence and multiplicity although when it was topically applied it was more effective. Moreover, a strong protective effect of genistein was found when UVB irradiation was combined either with DMBA or TPA [293]. Because UV irradiation induces ROS production, it is believed that the protective effect of genistein can be related to its antioxidant effect [294]. 

**Isoquercitrin** or quercetin-3-O-β-D-glucopyranoside, found similarly to other flavonoids in fruits, vegetables, and other plant-based foods and beverages, has been studied on several skin tumor cell lines (SK-MEL-2, SK-MEL-28, B16) and non-tumor cell lines (HaCaT) [295]. At a concentration of 25 µmol/L, it significantly inhibited the viability and clonogenicity of SK-MEL-2 cells and induced their death, but this effect was not observed in healthy cells. Apoptotic cell death was preceded by DNA fragmentation and cell cycle arrest in the G1/S phase of the cell cycle. This led to the activation of the mitochondrial apoptotic pathway, which was accompanied by a decrease in the anti-apoptotic protein Bcl-2, and an increase in pro-apoptotic proteins Bax, AIF, Endo G, and cleaved PARP. Caspases contribute to PARP cleavage. After isoquercitrin treatment, there was a reduction in inactive forms of caspases, pro-caspases -8 and -9. Another important mechanism is the modulation of the signaling pathways. The PI3K/Akt/mTOR transduction pathway has a significant impact on cell proliferation, survival, growth, cell cycle regulation, and cell death [296]. Isoquercitrin significantly reduced the levels of phosphorylated forms of all its members, indicating that the antiproliferative effect of IQ is also mediated by the suppression of the PI3K/Akt/mTOR signaling pathway [297].

**Isorhamnetin**, a derivative of quercetin (quercetin-3-methyl ether), has been shown to have a wide range of biological effects including anticancer [298]. In melanoma B16F10, cell, isorhamnetin dose-dependently inhibited proliferation and migration. Moreover, the exposition of cells to isorhamnetin resulted in apoptosis induction associated with the dysregulation of Bcl-2 family proteins and caspase-3 activation. Moreover, it inhibited the phosphorylation of Akt and suppressed the nuclear accumulation of NF-κB. In addition, animal experiments showed that isorhamnetin significantly decreased the volume of tumors initiated by B16F10 cell injection in C57BL/6 mice [299]. Another study showed the anticancer effect of isorhamnetin in non-melanoma cancer cells [300]. *In vitro*, isorhamnetin prevented EGF-induced neoplastic transformation of JB6 epidermal cells. Moreover, isorhamnetin inhibited the growth of A431 squamous carcinoma cells as well as the expression of COX-2, an important inflammatory mediator associated with skin carcinoma. In vivo, significant suppression of tumor volume and tumor weight in isorhamnetin-treated cells has been observed. Subsequent molecular analyses showed aberrant activation/deactivation of several signaling pathways such as ERK, Akt, PI3, and MEK in isorhamnetin-treated cells. Furthermore, Li and co-workers (2012) reported that quercetin-3-methyl ether suppressed the malignant transformation of mouse skin epidermal JB6 cells exposed to 12-O-tetradecanoylphorbol-13-acetate [301]. In addition, this chalcone inhibited the phosphorylation of ERK2 and induced cell cycle arrest at the G2/M phase. 

**Kaempferol**, a flavonoid found in several spices, is also a potent inhibitor of melanoma cell proliferation and survival. In the study by Qiang et al. (2021), kaempferol induced cell cycle arrest at the G2/M phase in murine melanoma cells followed by induction of apoptosis. Moreover, kaempferol also decreased the volumes and weights of tumors in mice injected with melanoma cells [302]. Other authors showed that kaempferol-induced apoptosis was associated with the downregulation of the m-TOR/PI3K/AKT pathway in human melanoma A375 cells [303]. Furthermore, kaempferol has been recently reported to inhibit melanoma metastasis due to the suppression of aerobic glycolysis of both human and mouse melanoma cells [304]. In another study, this flavonoid has been found to suppress UV-induced skin cancer. In experimental animals, local administration of kaempferol reduced tumor volume and incidence by 68% and 91%, respectively. As shown in detailed analyses, this effect has been associated with the inhibition of p90 ribosomal S6 kinase (RSK), mitogen and stress-activated protein kinase (MSK), and downstream molecules of the MAPK cascade [305]. In addition, the protective effect of kaempferol in UV-induced skin cancer can also be associated with the inhibition of Src kinase activity, which is an upstream MAPK regulator [306]. 

**Luteolin**, a flavonoid found in numerous fruits and vegetables, has been reported to possess a multi-targeted mechanism of anti-melanoma action. The research conducted by Schomberg et al. (2020) revealed that the growth inhibition of melanoma cells induced by luteolin was linked to the alteration of numerous genes involved in various pathways, including the extracellular matrix (ECM) pathway, oncogenic pathway, and immune response signaling [307]. Moreover, luteolin also showed a significant anticancer effect in nude mice injected with melanoma cells. In another study, luteolin suppressed melanoma cell proliferation and induced apoptosis via the inhibition of phosphorylation of AKT1 and PI3K, two pathways involved in the control of cell growth and survival. In addition, downregulated expression of MMP-2 and MMP-9 with concomitant increased expression of tissue inhibitors of metalloproteinases TIMP-1 and TIMP-2 has also been observed in the luteolin-treated A375 melanoma cells [308]. 

Epithelial–mesenchymal transition (EMT) plays an important role in cancer progression and metastasis. During cancer metastasis, tumor cells can undergo EMT, which allows them to detach from the primary tumor, invade surrounding tissues, enter the bloodstream or lymphatic system, and establish secondary tumors in distant organs. EMT is associated with downregulated expression of epithelial markers such as E-cadherin, while the expression of mesenchymal markers, such as N-cadherin, vimentin, and fibronectin, is upregulated [309]. The study reported by Li et al. (2019) indicated that the antiproliferative effect of luteolin in melanoma cells can be associated with decreased expression of proteins involved in hypoxia-inducible factor-1α/vascular endothelial growth factor (HIF-1α/VEGF) signaling pathway including HIF-1α, VEGF-A, and vascular endothelial growth factor receptor (VEGFR-2) [310]. In addition, decreased expression of MMP-2 and MMP-9 as well as *p*-Akt has also been found. Moreover, luteolin reversed EMT via N-cadherin, vimentin downregulation, and E-cadherin upregulation. 

Additionally, another mechanism leading to the suppression of proliferation and induction of melanoma cell death such as induction of ER stress, generation of reactive oxygen species (ROS), cell cycle arrest, inhibition of β3 integrin or signal transducers, and activators of transcription 3 (STAT3) signal pathways has also been reported [311,312,313,314]. 

It is well known that UVB-induced skin inflammation and oxidative stress contribute to the development and progression of skin cancer [315]. Luteolin exhibited the capacity to impede both UVB-triggered skin erythema and the elevation of cyclooxygenase-2, along with the production of prostaglandin E2 in the skin of volunteers. This effect was achieved through the disruption of the MAPK pathway [316].

**Myricetin** is a naturally occurring flavonoid with a wide distribution in various food sources. As a flavonol, it is predominantly found in onions, red wine, tea, berries, fruits, vegetables, medicinal herbs, and spices. In berries, vegetables, and fruits, myricetin is commonly present in the form of glycosides rather than free aglycones. Notably, the skins of several fruits, like red grapes, contain notably high levels of myricetin [317]. Myricetin exhibits strong anticancer activity and demonstrates promising therapeutic potential by targeting and regulating the expression of several molecular targets involved in critical processes such as cell proliferation, invasion, and metastasis as well as inflammation, apoptosis, and angiogenesis [318]. Jung et al. (2008) show that myricetin reduced UVB-induced COX-2 expression in mouse skin epidermal cells and this effect has been associated with the inhibition of Fyn kinase activity which plays a crucial role in the development of skin cancers [319]. Moreover, myricetin also significantly suppressed skin tumor incidence in animals exposed to UVB in a dose-dependent manner. In addition, the antiproliferative activity of myricetin against A431 human skin cancer cell lines was also evaluated. Myricetin suppressed A431 cell proliferation and decreased their colony-forming capacity. Moreover, it induced apoptosis associated with changes in the Bcl2/Bax ratio, increased production of ROS, and by reducing mitochondrial membrane potential. In addition, myricetin significantly inhibited the migration and invasion of skin cancer cells [320]. Furthermore, it was found that myricetin inhibited TPA-induced COX-2 expression in JB6 P+ cells by modulation of NF-кB activation [321]. In another study, these authors found that myricetin inhibited TPA- or EGF-induced transformation of JB6 P+ mouse epidermal cells together with the inhibition of mitogen-activated protein kinase (MEK), a kinase involved in tumor development. This effect was associated with the inhibition of ERK and RSK phosphorylation [322]. Later, Kumamoto et al. (2009) showed that EGF-induced JB6 P+ transformation can be efficiently inhibited by myricetin via the inhibition of Janus Kinase (JAK) 1 phosphorylation and, to a lesser extent, via STAT3 inhibition [323]. In addition, myricetin has been documented to, *in vivo*, in mouse skin, exert a suppressive effect on UVB-induced angiogenesis by modulating the activity of PI-3 kinase [324].

**Naringenin**, another flavonoid present in several spices, has been reported to possess an antiproliferative effect in a melanoma model. In a recent study, naringenin in SK-MEL-28 human and B16F10 murine melanoma cells induced apoptosis associated with activation of caspase-3 and PARP cleavage. Furthermore, in naringenin-treated melanoma cells, suppression of migration with a concomitant decrease of ERK1/2 and c-Jun N-terminal kinase (JNK) phosphorylation was observed. In addition, naringenin also inhibited angiogenesis as documented by the suppression of human umbilical vein endothelial cells (HUVECs) migration, inhibition of tube formation of HUVECs, as well as microvessels sprouting from rat aorta [325]. Another study showed that the antiangiogenic effect of naringenin can be associated with the inhibition of Two-Pore Channel 2 activity, an intracellular Ca^2+^ channel involved in several pathophysiological processes such as cancer cell proliferation, metastasis, or angiogenesis [326,327]. In the study by Lentiny and co-workers (2007), the anti-metastatic potential of naringenin has been evaluated [328]. Oral administration of naringenin to C57BL6/N mice bearing B16F10 cells significantly decreased the number of lung metastases by approximately 69% compared to untreated animals. Furthermore, in the study by Ahamad et al. (2014), naringenin has been reported to inhibit proliferation and induce apoptosis also in non-melanoma human epidermoid carcinoma A431 cells [329]. DNA fragmentation, ROS generation, mitochondrial depolarization, and caspase activation were associated with apoptosis in naringenin-treated cells. Another study showed the chemopreventive effect of naringenin in chemically induced skin cancers. In a two-stage skin carcinogenesis model (DMBA as inductor and croton oil as a promoter), treatment of experimental mice with naringenin significantly decreased the number, incidence, and size of skin papillomas [330]. As was mentioned above, inflammation and oxidative stress play a crucial role in UVB-induced skin carcinogenesis. Naringenin has shown potential in mitigating skin inflammation. In the study conducted by Martinez and co-workers [331] intraperitoneal application of naringenin to hairless mice reduced skin inflammation as documented by decreased skin edema, neutrophil accumulation, decreased activity of MMP-9, and suppressed production of numerous pro-inflammatory cytokines (e.g TNF-α, IFN-γ, and several interleukins). Moreover, it also decreased levels of superoxide anions. Later, these authors showed the protective effect of naringenin also after topical application [332]. Consistent with earlier findings, the protective effects of naringenin were associated with inhibition of skin edema and cytokines production. Apart from its role in suppressing superoxide anions, naringenin also acted to inhibit the generation of lipid peroxides. Additionally, naringenin upregulated the mRNA expression of key components in the antioxidant defense system, namely glutathione peroxidase 1, glutathione reductase, and the transcription factor Nrf2.

**Quercetin**, a natural flavonoid found abundantly in vegetables and fruits, is believed to have a variety of health benefits, including anti-oxidative, anti-inflammatory, antimicrobial, cardioprotective, antidiabetic, or anticancer properties, as well as its ability to boost the immune system [333]. Regarding skin cancers, quercetin’s anticancer effects can be attributed to several potential mechanisms. Quercetin has been found to inhibit proliferation and induce apoptosis in both melanoma and non-melanoma cells. It exerts these effects by modulating various mechanisms such as cell cycle arrest, downregulation of the anti-apoptotic proteins, and upregulation of the pro-apoptotic protein expression or caspase activation [334,335,336]. In addition to these effects, quercetin has been reported to modulate the activity of several signaling pathways associated with cell survival, growth, migration, malignant transformation, or cell death. Cao et al. (2014) demonstrated that the anticancer effect of quercetin can be associated with the inhibition of STAT3 phosphorylation and the subsequent suppression of STAT3 nuclear localization [337]. Moreover, quercetin also decreased STAT3 downstream gene expression including MMP-2, MMP-9, and VEGF. According to Kim et al. (2019), the mechanism of the antiproliferative effect of quercetin in human A375SM melanoma cells may include the regulation of the MAPK signaling pathway [338]. Moreover, significantly increased JNK and p38 phosphorylation has been observed in the tumors from mice transplanted with A375SM melanoma cells. In another study, the exposition of murine melanoma B16-BL6 cells to quercetin resulted in the significant inhibition of the cell invasion and migration associated with the suppressed gelatinolytic activity of MMP-9 and PKC signaling [339]. Some other studies showed that quercetin is capable of decreasing the metastatic potential of melanoma cells also via suppression of EMT. In the study conducted by Lin et al. (2011), it was observed that treating the highly invasive human epidermal carcinoma A431-III cells with quercetin caused a transformation of their mesenchymal-like morphology into an epithelial-like morphology [340]. Additionally, the treatment with quercetin resulted in the downregulation of mesenchymal markers such as N-cadherin, vimentin, fibronectin, Twist, and Snail while simultaneously upregulating the expression of the epithelial marker E-cadherin. Later, Patel and Sharma (2016) reported that quercetin in SK-MEL-28 human melanoma reversed collagen I-induced EMT via suppression of the expression of mesenchymal markers including vimentin, N-cadherin, Twist, and SNAIL [341]. Conversely, quercetin also increased the expression of epithelial markers such as VCAM1, ICAM1, and E-cadherin. Overall, these findings indicated that quercetin had the ability to reverse the EMT process and promote the restoration of an epithelial phenotype in melanoma cells. 

In addition, the association between the antiproliferative effect of quercetin and the modulation of signaling pathways such as Src/Stat3/S100A7 signaling [342], hepatocyte growth factor/c-Met signaling [343], epidermal growth factor signaling [344], phosphoinositide 3-kinase signaling [345], or retinoic acid-inducible gene I signaling [346] was observed.

In another study, quercetin was reported to be effective in a DMBA/TPA two-stage model of mouse skin carcinogenesis. The application of quercetin in a diet for 20 weeks significantly delayed the incidence of as well as decreased tumor multiplicity. Simultaneously, *in vitro* experiments using skin papilloma cells showed that quercetin suppressed insulin-like growth factor (IGF)-1 signaling. As proposed by the author, this mechanism might also play a role in hindering the progression of skin cancer in animals [347]. 

The extensive range of biological effects exhibited by quercetin has attracted significant interest. However, its limited skin permeation and deposition efficiency have hindered its potential application for skin cancer prevention. As a result, there is a need for a suitable carrier for quercetin to facilitate its efficient application onto the skin. Vincentini et al. (2008) used water/oil (w/o) emulsion as a topical carrier system for quercetin delivery. They found that w/o microemulsion significantly increased quercetin skin penetration in hairless mice. Furthermore, they observed that quercetin in w/o microemulsion prevented UVB-induced decrease in GSH levels and secretion/activity of metalloproteinases [348,349]. Later, Liu et al. used deformable liposomes to increase skin permeability of the quercetin. Remarkably higher penetration was observed for quercetin encapsulated within liposomes in comparison to quercetin in its un-encapsulated form. Moreover, histopathological assessment demonstrated superior protective attributes of quercetin-loaded liposomes compared to quercetin administered on its own. Subsequently, Zhu et al. [350] employed PLGA-TPGS nanoparticles to enhance the permeation and deposition of quercetin. Their study revealed that this combination augmented the protective effect of quercetin against UVB-induced alterations in mice skin. This was evidenced by the protection of collagen fibers, reduction in COX-2 expression, and inhibition of NF-kB activation. A similar enhanced, protective effect of quercetin was reported by Nan et al. [351]. In their study, they utilized chitosan nanoparticles loaded with quercetin, resulting in heightened permeation of the compound. Furthermore, in UVB-irradiated mice, the quercetin-loaded chitosan nanoparticles exhibited notable inhibition of the NF-kB/COX-2 signaling pathway compared to quercetin administered alone. Additionally, the loaded quercetin more effectively mitigated the deterioration of skin collagen structure and the occurrence of skin edema.

**Rutin**, a flavonoid found in many plants, has a broad spectrum of biological activities such as anti-inflammatory, anti-oxidative, neuroprotective, anti-nociceptive, anticancer and many other properties [352]. In human melanoma RPMI-7951 and SK-MEL-28 cell lines, rutin decreased viability, caused loss of adherence, and initiated changes in cell morphology. Moreover, rutin induced nuclear fragmentation, and chromatin condensation indicating apoptosis induction. In addition, rutin has been found to induce senescence in melanoma cells as confirmed by increased expression of beta-galactosidase, a protein associated with cell senescence [353]. Furthermore, in the study by Martínez Conesa et al., the anti-metastatic potential of rutin has been evaluated [274]. Results of the study showed the reduction of lung metastasis in rutin-treated animals injected with B16F10 melanoma cells. The ability of rutin to inhibit the lung tumor nodule induced by B16F10 melanoma cells’ injection in experimental animals has also been documented by Menon et al. [354]. Another study showed the potential of rutin to inhibit UVB-induced inflammation of hairless mouse skin. Topical application of rutin before UVB radiation significantly reduced UVB-induced expression of several pro-inflammatory mediators such as COX-2 and iNOS, and it prevented the activation of signaling pathways including STAT3, p38 MAPK, and JNK as well [355].

**Vitexin** is a flavonoid found in various plants, particularly in fruits, vegetables, and certain medicinal herbs and spices. It is often recognized for its potential health benefits due to its antioxidant and anti-inflammatory properties. Regarding skin cancers, in both extract and pure substance forms, vitexin has demonstrated the remarkable ability to effectively inhibit the growth, proliferation, and ability to form colonies of various melanoma cell lines, including mouse melanoma cell B16F10 and human malignant melanoma cell lines A375, and vemurafenib-resistant melanoma cells (A375, Sk-Mel-5, and Sk-Mel-28). In addition, vitexin induces a block at the G2/M phase of the cell cycle and apoptosis. In vitexin-treated cells, downregulation of the anti-apoptotic protein Bcl-2 and upregulation of the pro-apoptotic protein Bax have been observed followed by cleavage of the DNA repair enzyme PARP. The antiproliferative activity of vitexin was associated with DNA damage induced by increased levels of ROS and oxidative stress. At the protein level, phosphorylated ATM and ATR, protein kinases that were activated in response to DNA damage, were significantly upregulated. In addition, checkpoint kinase Chk2 was activated, levels of tumor suppressor proteins p21 and p53 were increased, and the accumulation of γH2AX, which interacts with kinases such as ATM and ATR, occurred. These results are consistent with *in vivo* experiments on xenografted mice, where vitexin at doses of 40 and 80 mg/kg resulted in a significant reduction in tumor growth [356]. As reported by Zhang et al. (2020), vitexin in human melanoma A375 and C8161 cells inhibited the expression of MMP-2 and MMP-9 as well as the expression of vimentin and Slug and Twist, a transcription factor involved in EMT [357]. In addition, vitexin also decreased the phosphorylation of Src, Janus Kinase (JAK)1, and JAK2 kinases followed by the deactivation of STAT3 signaling.

Molecular and cellular mechanisms of the antiproliferative action of flavonoids discussed in this review are listed in Table 3.

### 5.3. Other Phenolic Compounds

The chemical structures of selected other phenolic compounds discussed in this review are shown in Figure 4.

**1′-acetoxychavicol acetate** (ACA), derived from A. galanga, has been identified as the primary compound with a range of biological activities [71]. Among others, ACA has been evaluated as a chemopreventive agent in experimental skin cancers. Murakami et al. (1996) found that ACA significantly reduced TPA-induced skin cancers in female JCR mice [358]. In addition, ACA was found to be an inhibitor of reactive oxygen species’ (ROS’) production and a suppressor of lipid hydroperoxide formation. Another study confirmed the chemopreventive effect of ACA in TPA-induced skin carcinogenesis and this activity was associated with the inhibition of NF-κB activation [359].

***trans*-Anethole** is classified as a phenylpropene and is found in various plants, particularly in spices and herbs. It has been demonstrated that *trans*-anethole has excellent antibacterial, antifungal, antioxidant, and anti-inflammatory effects and it is “generally recognized as safe” and is, therefore, a valuable substance for use in medicine and skincare [171]. Its antiproliferative activity has been demonstrated on CRL-6475 tumor cells, where it exhibited significant selectivity compared to healthy human epidermal melanocytes (HEMa-LP) [360]. In concentrations of 3–12 µmol/L, *trans*-anethole inhibited the proliferation and clonogenicity of tumor cells. Apoptosis was detected using double staining with AO/EB and Annexin V/PI, and there was an increased population of dead cells. Furthermore, this phenylpropanoid significantly upregulated the expression of miR-498. When the cells were exposed to the miR-498 inhibitor, the antiproliferative effect of anethole was attenuated, demonstrating its critical role in cell proliferation, invasiveness, and metastasis [361,362]. The authors assume that this effect is caused by the modulation of the miR-498/STAT4 axis. The pro-apoptotic effects of alcoholic star anise extract, rich in *trans*-anethole, were demonstrated in several tumor cell lines, including the human melanoma cell line M14WM. They found a dose-dependent inhibition of proliferation (10, 25, and 50 μg/mL) and morphological changes typical of apoptosis [363].

**Arbutin**, a hydroquinone glucoside detected in many plants, is capable of inhibiting melanin production by inhibiting tyrosinase. In addition to this effect, arbutin has been proven to be an active anticancer agent against various types of cancer, such as bladder, breast, brain, cervical, colon, stomach, and other types of cancer [364]. The proapoptotic effect of arbutin and its acetyl derivative was studied in B16 melanoma cells. After cell cycle arrest in the G1 phase, apoptosis occurred, which was associated with the induction of mitochondrial dysfunction. Mitochondrial damage induced by arbutin and its acetyl derivative led to the loss of MMP and changes in the expression of Bcl-2 proteins, where Bcl-2 and Bcl-xL were downregulated [365]. Its antiproliferative effects were also confirmed by proteomic analysis in A375 cells (human malignant melanoma cells). Upregulated proteins included p53, VDAC-1, and 14-3-3G, while others were downregulated, such as vimentin, HSP90, and others [366].

**Capsaicin** (8-methyl-N-vanillyl-6-nonenamide), the major active component of chili peppers, is one of the most commonly used spices in the world. It is responsible for the pungent and spicy flavor of chili peppers. It is known to activate the transient receptor potential vanilloid 1 (TRPV1) receptor, which is involved in the perception of pain and heat, so when capsaicin comes into contact with the skin or mucous membranes, it can cause a burning or stinging sensation [367]. Apart from its sensory effects, capsaicin has been studied for its potential health benefits. It has shown anti-inflammatory properties and may have analgesic (pain-relieving) effects. Some studies suggest that capsaicin may also have antioxidant and anticancer properties [368]. Regarding skin cancers, some studies have suggested that capsaicin may have anticancer effects in melanoma. Capsaicin inhibited cell growth and induced apoptosis in human melanoma A375 cells. These effects have been associated with increased generation of nitric oxide (NO•) leading to p53 activation and subsequent activation of both mitochondrial as well as extrinsic apoptotic pathways [369]. Other authors reported that capsaicin in highly metastatic B16-F10 mouse melanoma cells suppressed migration via inhibition of the PI3-K/Akt/Rac1 signal pathway. A detailed study showed that capsaicin significantly decreased p85 phosphorylation, a regulatory subunit of PI3-K [370]. Another study showed that capsaicin in this type of melanoma cells induced apoptosis due to modulation of Bcl-2 family proteins expression, PARP cleavage, and caspase-3 activation [371]. A recent study by Chu et al. (2019) confirmed the significant antiproliferative effect of capsaicin on melanoma cells [372]. Using A375 and C8161 human melanoma cell lines showed the pro-apoptotic effect of this compound associated with PARP cleavage and caspase-3 activation. Additionally, the study revealed that capsaicin also triggered autophagy, as evidenced by the formation of autophagosomes and the accumulation of autophagy markers such as beclin1 and LC3 proteins. However, the inhibition of autophagy by 3-MA significantly enhanced the antiproliferative effect of capsaicin. This suggests that capsaicin-induced autophagy acts as a pro-survival process in these melanoma cells. Furthermore, a recent study revealed that the anti-melanoma effect of capsaicin is also associated with the modulation of the tumor-associated NADH oxidase (tNOX). Capsaicin directly inhibited tNOX activity and reduced tNOX expression on both protein and genetic levels. In addition, capsaicin in melanoma cells induced ROS generation followed by autophagy but not apoptosis induction. Moreover, capsaicin in female C57BL/6 mice bearing B16F10 melanoma cells significantly reduced tumor volume and tumor weight. In addition, protein analysis of the tumor tissue showed that capsaicin decreased tNOX expression also *in vivo* [373].

**Carnosic acid** (CrA) is classified as a phenolic diterpene compound and is predominantly present in plants of the *Lamiaceae* family, which encompasses notable herbs like rosemary, sage, and thyme. Rosemary, in particular, is widely recognized as a prominent natural source of carnosic acid It is known for its biological activities including antioxidants, anti-inflammatory, and neuroprotective properties. In addition, the anticancer effect of CrA has been intensively studied in both solid cancers and hematological malignities [374]. Regarding skin cancers, CrA has been reported to inhibit the growth, migration, and proliferation of B16F10 melanoma cells. Moreover, it blocked cell cycle arrest at the G0/G1 phase, upregulated p21, and downregulated p27. In addition, a combination study showed that CrA potentiated the effect of “classical” chemotherapeutic agents lomustine and carmustine. Finally, CrA decreased tumor growth in mice injected with B16F10 melanoma cells [375]. In another study, carnosic acid inhibited the migration and adhesion of B16F10 cells. Moreover, it inhibited the secretion of MMP-9, urokinase plasminogen activator (uPA), and vascular cell adhesion molecule (VCAM)-1. In addition, CrA was observed to promote the expression of E-cadherin while reducing the expression of vimentin and N-cadherin, which are markers associated with a mesenchymal phenotype. Based on these results, it is suggested that the inhibition of EMT might be a key factor contributing to the reduced cell migration in B16F10 cells treated with CrA [376]. In a recent study by Alcaraz et al. (2022), CrA was identified as a potent radiosensitizing agent [377]. The research demonstrated that when B16F10 melanoma cells were exposed to X-ray irradiation, CrA significantly reduced their cell survival, indicating its potential to enhance the effectiveness of radiotherapy in treating melanoma. Furthermore, another study showed the potential of CrA to enhance the cytotoxicity of β-lapachone in melanoma cells. The mechanism of β-lapachone cytotoxicity is due to the production of ROS via NAD(P)H quinone oxidoreductase 1 (NQO1) activity. Carnosic acid was able to induce the activity of NQO1 and enhanced the cytotoxicity of β-lapachone in melanoma cells [378].

**Carnosol**, a phenolic diterpene found mostly in rosemary, gained attention due to its potential health benefits and therapeutic properties. It exhibits antioxidant, anti-inflammatory, and anticancer properties, among others. Regarding its anticancer effects, studies have shown that carnosol may exhibit anticancer effects by interfering with various processes involved in cancer development and progression. It has been found to inhibit the growth of cancer cells, induce apoptosis, and suppress the formation of blood vessels that support tumor growth [379,380]. Furthermore, several studies also reported the preventive effect of carnosol against skin cancers. In the study by Huang et al. (2005), carnosol was found to inhibit the invasivity of highly metastatic mouse melanoma B16/F10 cells [381]. Treatments of cells with carnosol resulted in the inhibition of B16/F10 cell migration and invasivity. This effect was associated with decreased activity and expression of MMP-9. In addition, carnosol inhibited the activation of several signaling pathways, including ERK 1/2, AKT, p38, and JNK as well as inhibiting the NF-kappaB activity. Another author reported that the anti-proliferative and pro-apoptotic effect of carnosol in human melanoma G361 cells is associated with a marked elevation of ROS level. Detailed analyses showed that carnosol modulated the expression of several Bcl-2 family proteins, increased the cellular level of p53, and, on the other hand, prevented the activation of Src and STAT3. Subsequently, the expression of STAT3-dependent genes such as D-cyclins and survivin has also been downregulated [382]. Another study showed that carnosol prevented UVB-induced ROS production in human keratinocyte HaCaT cells resulting in a DNA damage decrease as documented by suppressed phosphorylation of γH2AX and Chk1, two markers of DNA damage. Moreover, carnosol prevented UVB-induced cell transformation into cancerous cells [383]. Another study showed that topical application of carnosol decreased the number of chemically induced (DMBA/TPA model) skin cancers in experimental animals. In addition, carnosol inhibited TPA-induced skin inflammation and the activity of ornithine decarboxylase [384]. Recently, Yeo et al. (2019) reported the anti-inflammatory effect of carnosol in UVB-irradiated animals [385]. Topical application of carnosol inhibited erythema, epidermal thickness, decreased inflammatory markers, iNOS, and COX-2. Furthermore, carnosol prevented STAT3 activation as well as decreased the expression of proinflammatory cytokines including TNF-α and IL-1β.

**Carvacrol**, a phenolic monoterpenoid that can be isolated from various plants including oregano or thyme, exhibits a diverse array of bioactivities that have proven valuable in clinical applications. These include antimicrobial properties, anti-inflammatory, and antioxidant effects as well as anticancer activity [386,387,388,389]. Regarding melanoma, in carvacrol-treated human melanoma A375 cancer cells, significant suppression of cell growth and induction of apoptosis has been reported. Flow cytometric analyses showed cell cycle arrest at the G2/M phase in carvacrol-treated cells and an increased number of cells with sub-G0 DNA content, which is considered a marker of apoptosis. Western blot analyses revealed a decrease in Bcl-2 protein expression and PARP cleavage [390]. Another study showed that carvacrol entrapped in chitosan nanoparticles was significantly more potent against A-375 melanoma cells compared to carvacrol only [391]. Recently, Nanni et al. (2020) reported a strong anti-melanoma oregano extract [392]. An oregano extract in murine and human melanoma cells induced ROS production and inhibited melanoma cell proliferation. Moreover, it triggered apoptosis and necroptosis via DNA and mitochondria damage. On the other hand, no cytotoxicity of oregano extract was observed in non-cancer cells.

**Cinnamaldehyde** is an organic compound that gives cinnamon its characteristic flavor and aroma and it is widely used as a flavoring agent in food and beverages due to its sweet, spicy, and warm taste. Apart from its culinary applications, cinnamaldehyde has gained attention for its potential health benefits. Studies have suggested that it may have antioxidant, anti-inflammatory, and anticancer properties [393,394,395].

Regarding melanoma, cinnamaldehyde has been reported to impair melanoma cell viability, proliferation, and invasiveness, as well as tumor growth. In the study by Cabello et al. (2009), the exposition of human melanoma A375, G361, and LOX cells to cinnamaldehyde led to decreased melanoma cell proliferation associated with cell cycle arrest at the G1 phase [396]. Further analyses showed that cinnamaldehyde induced apoptosis as confirmed by annexin-V/propidium iodide staining and activation of caspase-3. At sub-apoptotic concentrations, cinnamaldehyde inhibited cell migration. Furthermore, the application of cinnamaldehyde on A375 human melanoma cells resulted in an increase in the generation of ROS, along with the activation of genes involved in the cellular response to oxidative stress, including heme oxygenase-1, sulfiredoxin 1 homolog, and thioredoxin reductase 1. Moreover, daily application of cinnamaldehyde to mice bearing human A375 melanoma xenograft led to a decrease in tumor volume and weight when compared to untreated controls. Another study showed that the application of cinnamaldehyde to mice injected with melanoma cells led to significant suppression of tumor growth and decreased density of new blood vessels in the tumor. Western blot analyses showed downregulation of HIF-α and VEGF expression indicating that cinnamaldehyde-induced inhibition of angiogenesis can contribute to its anti-melanoma effect [397]. A study conducted by Patra et al. (2019) uncovered that the antiangiogenic effect of cinnamaldehyde is linked to the PI3/Akt/mTOR pathway, which is involved in the regulation of the transcription and translation of HIF-1α [398]. Moreover, at the molecular level, cinnamaldehyde decreased VEGF secretion, phosphorylation of VEGF receptor, and activity of MMP-2 and MMP-9. In vivo experiments showed that cinnamaldehyde suppressed invasion and metastasis in mice bearing B16F10 melanoma cells xenograft.

**Curcumin**, a diarylheptanoid belonging to the group of curcuminoids, is the primary bioactive compound found in turmeric, giving it its characteristic bright yellow-orange color. Vogel and Pelletier originally extracted curcumin from the rhizomes of C. longa in 1815 [200]. Curcumin’s impact on melanoma cell survival and proliferation was examined in a number of *in vitro* experiments that documented its pleiotropic mechanism of action. Zhang et al. (2015) studied the antiproliferative effect of curcumin using A375 human melanoma cells [399]. They found that curcumin decreased melanoma cell proliferation, migration, and invasion. Moreover, western blot analyses showed decreased expression of MMP-2 and MMP-9. Curcumin also decreased phosphorylation of JAK and STAT3 proteins followed by induction of apoptosis. Another study showed that the pro-apoptotic effect of curcumin is associated with the suppression of the transcriptional activity of NF-кB. In murine melanoma cells exposed to curcumin, Marín et al. (2007) documented the inhibition of melanoma cell proliferation, viability, and apoptosis induction [400]. A further study showed inhibition of NF-кB activity and decreased expression of NF-кB-targeted genes including COX-2 and cyclin D1. The association between apoptosis induction and modulation of NF-кB activity in curcumin-treated melanoma cells has also been referred by Siwak et al. (2005) and Zheng et al. (2004) [401,402]. Furthermore, in the study conducted by Jiang et al. (2014), curcumin has been reported to initiate the mitochondrial pathway of apoptosis in human melanoma cell lines [403]. They found that curcumin stimulated the expression of pro-apoptotic Bax protein and vice versa, inhibiting the expression of antiapoptotic Mcl-1 and Bcl-2 proteins. Apoptosis has also been associated with decreased levels of NF-κB p65 protein resulting in inhibition of NF-кB signaling, modulation of p38, and p53 activity, as well as with DNA damage. Interestingly, curcumin also activated caspase-8 and caspase-3 indicating that the extrinsic apoptosis pathway may also be activated in curcumin-treated melanoma cells. The same results i.e., activation of caspase-8 and caspase-3 were reported by Bush et al. (2001) who also found that curcumin caused aggregation of the Fas receptor and these effects were p53 and Bcl-2 proteins independent. In another study, curcumin-induced apoptosis has been associated with curcumin-induced oxidative stress in human melanoma A357 cells [404]. Curcumin increased the generation of ROS and, on the other hand, decreased intracellular GSH levels resulting in apoptosis initiation [405]. ROS-induced cell death in curcumin-treated mouse melanoma cells has also been reported in the study conducted by Kocyigit and Guler (2017) [406]. Curcumin in B16-F10 melanoma cells increased ROS levels, caused DNA damage, and decreased mitochondrial membrane potential (MMP) with subsequent apoptosis induction. In addition, numerous studies reported different mechanisms of curcumin-induced growth or survival suppression such as induction of ER stress [407], autophagy induction [408], or modulation of signaling pathways [205,399,409,410].

Additionally, several studies also confirmed an anticancer effect of curcumin *in vivo*. In mice bearing B16F10 melanoma cells xenograft, orally administered curcumin significantly decreased the number of lung metastasis and also prolonged the survival of experimental animals [354]. The ability of curcumin to prevent lung metastasis in mice injected with B16F10 melanoma cells was confirmed recently. Pulmonary administration (for better bioavailability) of curcumin-protected melanoma lung metastasis was completed in a dose-dependent manner. Histological analysis showed a decrease in the cell proliferation marker Ki-67 in curcumin-treated animals [411]. Furthermore, curcumin has been reported to inhibit skin tumors also in the mice bearing SCC cell line (SRB12-p9) xenograft. Curcumin applied topically or by oral gavage significantly reduced tumor growth and volume. Further analyses revealed that this effect was associated with the inhibition of AKT, pS6, 4EBP1, STAT3, and ERK1/2 phosphorylation [412]. In addition, curcumin also prevented skin cancer formation in UVB-irradiated animals. Both oral and topical applications of curcumin to UVB-irradiated SKH-1 hairless mice led to slower tumor onset and decreased tumor multiplicity when compared to untreated animals [413]. The protective effect of curcumin on UVB-induced skin tumorigenesis has also been reported by Tsai et al. [414]. Topically applied curcumin significantly decreased the number of tumors and tumor volume. Furthermore, curcumin significantly suppressed UVB-induced apoptosis and thymine dimers, a marker of UVB-induced DNA damage.

**Ellagic acid** (EA), is a naturally occurring polyphenolic compound found in various plant-based foods, particularly in fruits such as strawberries, raspberries, blackberries, pomegranates, and walnuts. It belongs to the class of hydrolyzable tannins and has been the subject of scientific research due to its potential health benefits. Some studies have investigated the effects of EA on melanoma. Human metastatic melanoma cell lines 1205Lu, WM852c, and A375 were incubated with different concentrations of EA for 72 h [415]. According to the study, EA was found to induce apoptosis and cause cell cycle arrest at the G1 phase, followed by apoptosis induction. Additionally, the study revealed that EA’s antiproliferative effects may be attributed to its ability to inhibit the NF-κB pathway. Other authors investigated the anticancer effects of EA on melanoma cells *in vitro* and *in vivo* [416]. The findings from their study demonstrated that EA notably suppressed the proliferation, migration, and invasion of WM115 and A375 melanoma cells, and these effects were achieved by targeting the EGFR signaling pathway. As the authors reported, incubation of WM115 and A375 melanoma cells with EA led to a reduction in EGFR expression. In addition, in A375-bearing animals, EA significantly reduced tumor size and weight, increased expression of E-cadherin and decreased EGFR phosphorylation. Furthermore, there are several reports documenting the protective effect of EA against UVB-induced damage to human keratinocytes. Hseu et al. (2012) reported that EA significantly decreased UVB-induced ROS generation and prolonged HaCaT viability with concomitant inhibition of UVB-induced apoptosis [417]. The preventive effect of EA correlated with increased expression of heme oxygenase 1 (HO-1) and SOD. This was followed by the downregulation of Keap1 and increased activation of Nrf2. Another author found that EA modulated the activity of cytokines in UVB-irradiated HaCaT cells. It decreased the expression of proinflammatory IL-6 and upregulated the expression of anti-inflammatory IL-10. However, other cytokines such as IL-6, IL-8, MCP-1 (monocyte chemoattractant protein), and TNF-α did not show significant changes in response to EA treatment [418].

**Eugenol** (4-allyl-2-methoxyphenol), a phenolic compound highly abundant in various spices such as cinnamon, basil, bay leaves, cloves, and nutmeg, demonstrates antioxidant, antibacterial, antiviral, anti-inflammatory, and anti-proliferative properties through diverse mechanisms [419,420,421]. Mechanisms of antiproliferative and apoptosis-inducing mechanisms have been reviewed in detail by Jaganthan and Supriyanto (2012) [422]. Regarding melanoma cells, eugenol has been reported to induce apoptosis in human melanoma G361 cells. Apoptosis was associated with the activation of caspase-3, PARP-cleavage, and DNA fragmentation [423]. Another study showed that eugenol inhibited different melanoma cells in a time-dependent manner. Moreover, it blocked the cell cycle at the G1 phase and induced apoptosis associated with blebbing of membranes, chromatin condensation, and cytoplasm shrinking. In addition, it was also found that eugenol modulated the expression of the members of the E2F family of transcription factors and it is suggested that via this effect eugenol can inhibit cell proliferation and cell cycle, and induce apoptosis. Furthermore, eugenol significantly decreased tumor size and decreased the development of metastasis in B16F10 xenograft mice [424]. The ability of eugenol to cause cell cycle arrest at the G1 phase has also been reported by Ra Choi et al. [425]. G1 arrest in G361 melanoma cells exposed to eugenol has been associated with a decrease in the expression of cycle-related molecules including cyclins A, D3, and E, and cyclin-dependent kinases cdk2, cdk4, and cdc2. On the other hand, in the study conducted by Júnior et al. in 2016, eugenol-induced apoptosis in SK-Mel-28 and A2058 melanoma was accompanied by cell cycle arrest at the G2/M phase indicating that the effect of eugenol can be cell type-dependent [426]. Moreover, the study conducted by Valizahed et al. (2021) indicated that the nanoformulation of eugenol exhibit improved chemotherapeutic effectiveness [427]. Chitosan nanoparticles loaded with eugenol demonstrated a significant effect against human melanoma cell lines (A-375) compared to eugenol in its non-formulated state.

In addition to melanomas, eugenol has also been reported to restrict experimentally induced skin cancers. In the study by Pal et al., skin cancers have been induced by the application of DMBA as an initiator and croton oil as a promoter [428]. In the untreated control animals, the tumor incidence was 100% while in eugenol-treated cells the number of mice with skin tumors was significantly reduced (42%). Furthermore, the size of tumors in the eugenol group was significantly smaller and the overall survival in eugenol-treated animals was significantly increased. In addition, eugenol significantly decreased cellular proliferation in skin tumors and this effect was associated with the suppression of expression of two oncogenes (c-Myc and H-ras) associated with cell proliferation. A similar effect of eugenol in the prevention of DMBA/croton oil-induced skin carcinogenesis is also reported by Sukumaran and co-workers (1994) [429]. A strong preventive effect of eugenol has also been observed in chemically induced skin cancer where TPA has been used as a tumor promoter instead of croton oil. Eugenol applicated before TPA significantly decreased cell proliferation, ornithine decarboxylase (ODC) activity, expression of COX-2, and iNOS, as well as the secretion of proinflammatory cytokines including IL-6, TNF-α, and PGE2. Moreover, in eugenol-treated animals, the suppression of NF-κB/p65 translocation has been detected [430]. 

In addition, an *in vitro* study in non-melanoma A431 cells (human epidermoid carcinoma) showed that eugenol induced the block of the cell cycle at the G0/G1 phase. Furthermore, it also induced the translocation of the aryl hydrocarbon receptor (AhR) to the nucleus, and this effect can be associated with the modulation of cell cycle regulatory proteins involved in the transition from the G0/G1 phase to the S phase [431]. 

**6-Gingerol**, together with other gingerols, is responsible for the pungent taste. Gingerols are the predominant compounds in fresh ginger rhizomes. Ginger contains several gingerols with different chain lengths, ranging from n6 to n10, with 6-gingerol being the most abundant among them [432]. 6-gingerol has been reported to possess several pharmacological activities such as antioxidant, anti-inflammation, anti-aggregatory or antimicrobial properties [433]. In addition, it has been studied for its potential anticancer properties. Studies conducted in cell cultures and animal models have shown that gingerol can inhibit the growth and proliferation of cancer cells. It has been found to induce apoptosis and suppress angiogenesis. Gingerol has also demonstrated the ability to modulate various signaling pathways involved in cancer development and progression [434,435,436]. Furthermore, 6-gingerol has also been studied as a potential anticancer agent against skin cancers. In the study by Nigam et al. (2009), 6-gingerol demonstrated significant cytotoxic effects against non-melanoma human epidermoid carcinoma A431 cells [437]. This effect was associated with the generation of ROS followed by a decrease in MMP and apoptosis induction. The findings also indicated that the perturbation in the mitochondrial membrane was associated with the disruption of the Bax/Bcl-2 ratio, both at the gene and protein levels. Another study showed that topically applied 6-gingerol inhibited COX-2 expression in mouse skin stimulated by the TPA. Furthermore, 6-gingerol suppressed the activity of NF-кB. In addition, it prevented TPA-induced activation of the p38 MAP kinase that regulates COX-2 expression in mouse skin [438]. Later, they reported that 6-gingerol *in vitro* reduced UVB-induced ROS production and caspase activation in human keratinocyte HaCaT cells. Moreover, it inhibited the UVB-induced activation of NF-кB. In addition, local application of 6-gingerol prior to UVB irradiation of hairless mice significantly suppressed UVB-induced COX-2 expression and NF-кB activation in mouse skin, indicating that 6-gingerol can be effective against UVB-induced skin damage [439]. In the study by Ju et al., 6-gingerol decreased tumor incidence, growth, and prolonged survival of mice inoculated with B16F1 melanoma cells [440]. Additionally, treatment with 6-gingerol resulted in a significant influx of tumor-infiltrating lymphocytes such as CD4 and CD8 T-cells, as well as B220+ B-cells, into the tumor microenvironment and this effect can play a role in the anticancer effect of 6-gingerol. Another study demonstrated that 6-gingerol exhibits antiangiogenic effects. In vitro experiments revealed that 6-gingerol inhibited the proliferation of HUVECs stimulated by VEGF and bFGF. It also suppressed the formation of capillary-like tubes induced by VEGF in different experimental models. Furthermore, in an *in vivo* study using mice implanted with B16F10 melanoma cells, treatment with 6-gingerol led to a reduction in the number of lung metastases. These findings suggest that 6-gingerol may have therapeutic potential in targeting angiogenesis and metastasis in cancer [436]. In a study by Park et al., topical application of 6-gingerol prevented TPA-induced inflammation in experimental animals and decreased the number of skin papillomas in the DMBA/TPA model of chemical skin carcinogenesis [441]. Additionally, it inhibited TPA-induced increased activity of epidermal ornithine decarboxylase, which plays an important role in skin carcinogenesis.

**6-shogaol** is another bioactive compound found in ginger. It is derived from gingerol during the drying or cooking process of ginger. It possesses unique properties and potential health benefits including anti-inflammatory, antioxidant, anticancer, and neuroprotective effects [129,442]. In the context of skin cancers, a study reported that 6-shogaol exhibited greater efficacy than 6-gingerol or curcumin in inhibiting tumor promotion induced by TPA in experimental animals. Topically applied to mouse skin, 6-shogaol significantly suppressed TPA-stimulated transcription of iNOS and COX-2 mRNA expression. Additionally, it was observed that 6-shogaol reduced the nuclear translocation of NF-кB, a protein involved in inflammatory responses and cell survival. Moreover, 6-shogaol demonstrated significant inhibition of signaling pathways associated with cell proliferation, survival, and inflammation including Erk1/2, p38 MAPK, PI3K/Akt, and JNK1/2. Furthermore, in a DMBA/TPA skin tumor model, 6-shogaol significantly reduced the occurrence of multiple skin tumors [443]. In a study conducted by Chen et al. (2019), it was found that 6-shogaol exhibited a protective effect against UVB-induced oxidative stress in HaCaT cells [444]. When HaCaT cells were exposed to UVB radiation without any treatment, it resulted in elevated levels of ROS and increased activation of p38, ERK, and JNK signaling. However, when HaCaT cells were treated with 6-shogaol prior to UVB irradiation, it significantly reduced ROS levels and prevented the activation of p38, ERK, and JNK. This suggests that 6-shogaol can mitigate the oxidative stress induced by UVB radiation by inhibiting the activation of the MAPK signaling pathway. Additionally, the study revealed that 6-shogaol prevented the depletion of the Nrf2 protein levels induced by UVB irradiation. Nrf2 plays a crucial role in cellular defense against oxidative stress. Therefore, the protective effect of 6-shogaol against oxidative stress can be partially attributed to its ability to prevent the depletion of Nrf2 protein levels.

**Thymol**, a phenolic monoterpene, is a major component of essential oils derived from thyme. It shares the same chemical formula as carvacrol, another compound found in thyme and oregano, but they differ in the position of the hydroxyl group on the benzene ring. While carvacrol has the hydroxyl group in the ortho position, thymol has it in the para position. Both thymol and carvacrol are aromatic compounds present in various plants belonging to the *Lamiaceae* and *Verbenaceae* families [445]. Thymol exhibits a wide range of biological activities, such as antioxidative, antimicrobial, antiviral, antihypertensive, immunomodulatory, and anticancer properties [446]. However, the research on the impact of thymol specifically on skin cancers is limited, with only a few studies conducted in this area. Thymol in B16-F10 melanoma cells exhibited moderate cytotoxicity and this effect was associated with the induction of oxidative stress. In addition, radical scavengers such as GSH, vitamin E, butylated hydroxyanisole, or butylated hydroxytoluene suppressed thymol-induced cytotoxicity in melanoma cells [447]. Another study showed that dry extract from *T. vulgaris* L. and thymol, its main active compound, prevented UVA- and UVB-induced damage of keratinocytes cell line NCTC 2544. Both treatments decreased UV-induced ROS production as well as DNA damage [448]. Later these authors documented the protective effect of *T. vulgaris* extract and thymol in UVA- and UVB-irradiated HaCaT (keratinocytes) cell line. Both compounds prevented UVA- and UVB-induced genotoxicity as well as the generation of ROS and apoptosis induction [449].

Another author reported that thymol decreased UVB-induced genotoxicity in ex vivo human skin models. Moreover, both compounds prevented UVB-induced cell toxicity as documented by decreased levels of lactate dehydrogenase (LDH) [450].

Molecular and cellular mechanisms of the antiproliferative action of other selected phenolic compounds are listed in Table 4.

## 6. Conclusions

Throughout history, spices have been utilized for both culinary and medicinal purposes. Their ability to enhance the taste, smell, and appearance of food and beverages is well known, but they also offer protection against various acute and chronic diseases. Recent research has highlighted the health properties of spices, particularly their bioactive constituents, with a focus on phenolic compounds.

Phenolic compounds have gained attention for their potential role in preventing and managing cancer. While they may not be as potent as conventional chemotherapeutic drugs, their capacity for cancer prevention is evident. In the case of skin cancer, utilizing phenolic compounds is particularly appealing due to their widespread availability, cost-effectiveness, and good level of tolerance. The skin’s accessibility makes it an advantageous area for both observation and topical treatment, encouraging research into the effectiveness of these natural compounds.

Studies have demonstrated that phenolic compounds exhibit anticancer effects through various mechanisms, such as antioxidant, proapoptotic, anti-inflammatory, anti-metastatic, and antiangiogenic actions, and their modulation of signaling pathways are related to cell life and death. Additionally, they may counteract the damaging effects of solar UV radiation and environmental carcinogens.

However, it is crucial to acknowledge that much of the current evidence comes from laboratory and animal studies. To establish phenolic compounds’ true effectiveness and safety in preventing or treating skin cancers, further clinical trials involving human subjects are essential. By conducting more research, we can better understand the potential benefits and limitations of using these natural compounds in the fight against skin cancer.

## Figures and Tables

**Figure 1 molecules-28-06251-f001:**
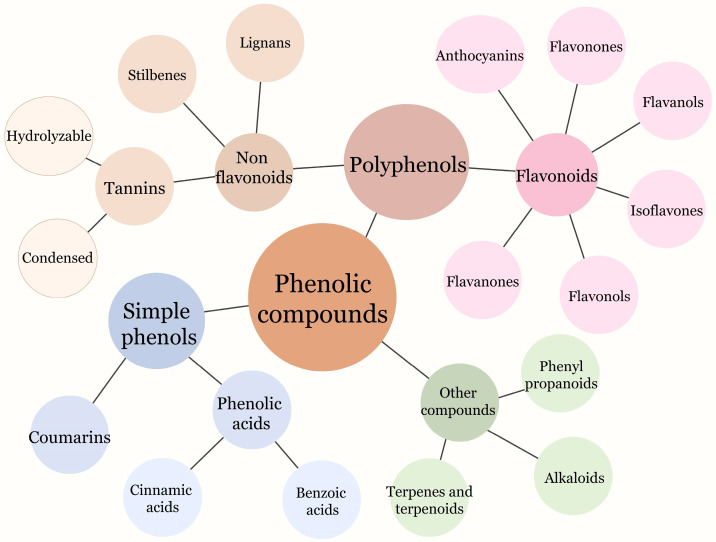
Classification of phenolic compounds.

**Figure 2 molecules-28-06251-f002:**
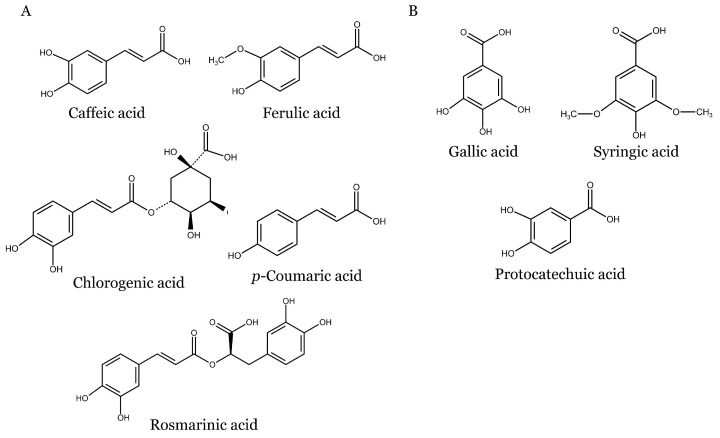
Structures of phenolic acids discussed in this review. (**A**)—hydroxycinnamic acid derivatives, (**B**)—hydroxybenzoic acid derivatives.

**Figure 3 molecules-28-06251-f003:**
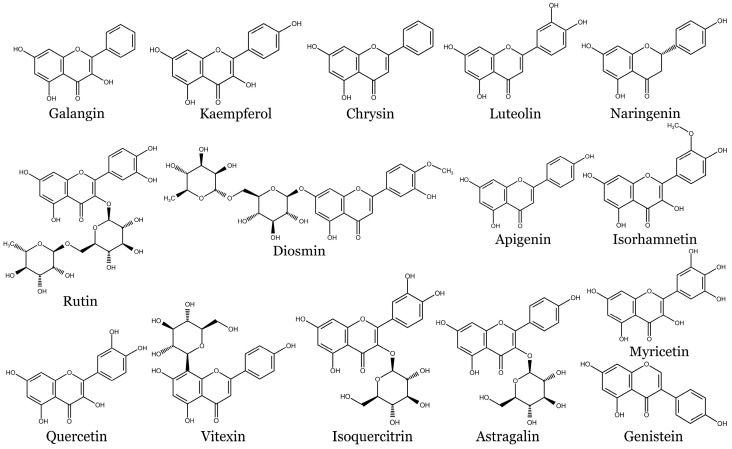
Structures of selected flavonoids discussed in this review.

**Figure 4 molecules-28-06251-f004:**
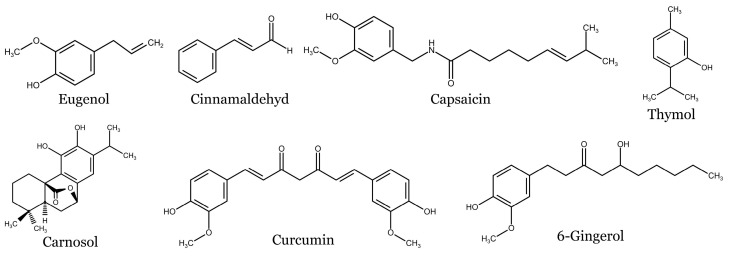
Structures of selected phenolic compounds discussed in this review.

**Table 1 molecules-28-06251-t001:** List of phenolic compounds isolated from discussed spices.

Spice	Phenolic Compounds	References
Allspice	Flavonoids: quercetin, quercitrin, kaempferol, and catechin, naringenin Phenolic acids: syringic acid, caffeic acid, coumaric acid, and cinnamic acid Phenylpropanoids: eugenol, methyl eugenol, isoeugenol, and chavicol Tannins: vascalaginone and grandininol	[65,66,67,68]
Alpinia galanga	Chalcones: galanganones A–C Flavonoids: galangin, galangin-3-methylether, and kaempferol, quercetin, isorhamnetin, apigenin, kumatakenin, and pinocembrin Lignans: galanganal and galanganol A–C Phenolic acids: ferulic acid and gallic acid Phenylpropanoids: 1′-acetoxychavicol acetate, and methyleugenol Tannins: ellagic acid	[72,73,74]
Black cumin	Flavonoids: kaempferol, kaempferol-3-glucoside, quercetin, quercitrin, and diosmin Monoterpenes: *p*-cymene Phenolic acids: caftaric acid, *p*-hydroxybenzoic acid, syringic acid, protocatechuic acid, and chlorogenic acid Phenylpropanoids: thymohydroquinone, thymol, carvacrol, and t-anethole	[81,82,83]
Black pepper	Alkaloids: piperine Terpenes: β-caryophyllene, limonene, β-pinene, α-pinene, sabinene, camphene, linalool, and terpinene-4-ol Phenolic acids: hydroxybenzoic acid, gallic acid, caffeic acid, and hydroxycinnamic acids Flavonoids: quercetin, catechin, epicatechin, myricetin kaempferol, isoquercetin, and isorhamnetin	[84,85,86]
Cinnamon	Phenylpropanoids: eugenol Phenolic acid: caffeic acid, chlorogenic acid, gallic, rosmarinic acid, *p*-coumaric acid, protocatechuic acid, *p*-hydroxybenzoic acid, and trans-vanillic acid Flavonoids: rutin, apigenin, catechin, and epicatechin Other compounds: cinnamaldehyde, cinnamyl alcohol, cinnamyl acetate, terpenes, and terpenoids	[97,98,99,100]
Coriander	Phenolic acids: caffeic acid, chlorogenic acid, ferulic acid, gallic acid, o-coumaric acid, trans-hydroxycinnamic acid, *p*-coumaric acid, rosmarinic acid, salicylic acid, trans-cinnamic acid, and vanillic acid Flavonoids: rutin, luteolin quercetin, kaempferol, naringin, apigenin, and diosmin Other compounds: esculin, esculetin, catechin, orientin, and maleic acid	[103,104,105,106,107]
Fenugreek	Flavonoids: quercetin, luteolin, vitexin, isovitexin, kaempferol, tricin, and naringenin Phenolic acid: *p*-coumaric acid, caffeic acid, and chlorogenic acid	[118,119,120,121]
Ginger	Flavonoids: quercetin Phenolic compounds: gingerols (6-gingerol, 8-gingerol, and 10-gingerol), shogalols (6-shogaol, 8-shogaol, and 10-shogaol), gingerenone-A, zingerone, 8-paradol, and 6-dehydrogingerdione	[128,129,130,131,132,133]
Oregano	Flavonoids: rutin, naringin, hesperetin, naringenin, apigenin, luteolin, acacetin, and vitexin Phenolic acids: caffeinic acid and rosmarinic acid Other compounds: carvacrol and thymol	[136,137]
Nutmeg	Phenolic acids: protocatechuic acid, caffeic acid, vanillic acid, *p*-coumaric acid, ferulic acid, and sinapic acid Flavonoids: catechin, epicatechin, rutin, quercitrin, isoquercitrin, quercetin, and kaempferol Other phenolic compounds: ellagic acid, myristicin, and elemicin	[146,147]
Red chili	Alkaloids: capsaicinoids (capsaicin, dihydrocapsaicin, nordihydrocapsaicin, homodihydrocapsaicin, and homocapsaicin) Flavonoids: quercetin, luteolin, kaempferol, myricetin, apigenin, naringenin, catechin, and epigallocatechin Phenolic acids: gallic acid, protocatechuic acid, vanillic acid, hydroxyl cinnamic acids, caffeic acid, ferulic acid, chlorogenic acid, and cinnamic acid	[149,150,151,152,153]
Rosemary	Phenolic acids: salvianic acid, caffeic acid, rosmarinic acid, and salvianic acid A Flavonoids: luteolin, luteolin−7-O-rutinoxide, luteolin-7-glucoronide, hesperidin, apigenin, cirsimaritin, genkwanin, and salvigenin Terpenes: rosmadial, 7-methylrosmanol, carnosol, carnosic acid, and 12-methylcarnosic acid	[158,159]
Saffron	Alkaloids: pyridin-3-ylmethanol, harman, and tribulusterine Phenolic acids: protocatechuic acid, 4-hydroxybenzoic acid, vanillic acid benzoic acid, and *p*-coumaric acid Flavonoids: kaempferol, kaempferide, kaempferol-3-O-sophoroside-7-O-glucoside, kaempferol-3,7,4′-triglucoside, kaempferol 7-O-β-D-glucopyranoside, isorhamnetin-3,4′-diglucoside, isorhamnetin-3-O-glucoside, astragalin, sophoraflavonoloside, and helichrysoside	[164]
Sichuan pepper	Polyphenols: isovitexin, vitexin, hyperoside, isoquercitrin, rutin, foeniculin, trifolin, quercitrin, astragalin, and afzelin	[169]
Star anise	Phenylpropanoids: cis- and trans-anethole, estragole, anisylacetone, ρ-anisaldehyde, and foeniculin	[177]
Sumac	Tannins: galloylhexose, O-galloylmorbergenin, and O-galloyl arbutin Flavonoids: quercetin, myrecetin, apigenin, kaempferol, and quercetin 3-glucoside Phenolic acids: gallic acid, ellagic acid, and protocatechuic acid Anthocyanins: delphidin-3-glucoside, cyanidin 3-(2″-galloyl)galactoside, and cyanidin-3-glucoside	[193]
Thyme	Monoterpenes and terpenoids: thymol and carvacrol Phenolic acids: caffeic acid, quinic acid, *p*-coumaric acid, ferulic acid, cinnamic acid, and rosmarinic acid Flavonoids: apigenin, naringnin, and luteolin-7-o-rutinose	[199]
Turmeric	Curcuminoids: curcumin, demethoxycurcumin, and bisdemethoxycurcumin Phenolic acids: gallic acid, protocatechuic acid, chlorogenic acid, cinnamic acid, and ferulic acid Flavonoids: rutin, myricetin, genistein, quercetin, catechin, and epicatechin	[202]

**Table 2 molecules-28-06251-t002:** Mechanisms of antiproliferative effect of phenolic acids based on *in vitro* studies. Arrows indicate an increase (↑) or decrease (↓) in the levels/activity of the molecules.

Phenolic Acids	Mechanism of Action	References
Caffeic acid	Inhibition of proliferation and colony formation, cell cycle arrest, induction of apoptosis, reduction of cell viability ↓ Erk1/2 signaling pathway, phospho- p90RSK, phospho-c-Myc, phospho-Elk1, COX-2, AP-1, Nf-кB, Fyn, DNA damage, ROS generation, PI3K, Akt, phospho-JNK, phospho-p38, ↑ PTEN, expression of caspase -1, -3, -8	[209,213,214,215]
Chlorogenic acid	Induction of apoptosis, inhibition of viability of tumor cells ↓ TGFβ, SOD1, SOD2, GPX1 ↑ activity of caspase -3	[216,217,218]
*p*-Coumaric acid	Inhibition of cell viability, proliferation and colony formation, S phase cell cycle arrest, induction of apoptosis, modulation of TME ↓ cyclins A and E, Cdk2, Bcl-2 ↑ cleaved caspase -3, -9, Apaf-1, cytochrome c release, Bax	[225,226]
Ferulic acid	Induction of apoptosis, reduction of colony formation, modulation of angiogenesis and inflammation, reduction of UVB-induced DNA damage ↓ MMP-2, MMP-9, iNOS, IL-6, TNF-α, VEGF, Bcl-2 ↑ Bax	[229]
Gallic acid	Induction of apoptosis, modulation of glycolysis, inhibition of migration and invasion, S phase cell cycle arrest, ROS generation ↓ MMP, MMP-2, Ras, Erk1/2, PI3K, p38-MAPK, Akt, Bcl-2, GADD153, Bcl-2, Bcl-xL, Mcl-1, ↑ activity capsase -3, -8, -9, Bid, cytochrome c release, GRP78, Fas, FasL, cleaved PARP, AIF, Bad, Bax, VDAC-1, SOD, G-3-P DHase, glucokinase, enolase, aldolase, ATPase	[232,233,234]
Protocatechuic acid	Modulation of inflammation and oxidative stress, inhibition of migration, invasion and metastasis ↓ MMP-2, Ras, phospho-Akt, nuclear NF-кB, PI3K, phospho-Akt, RhoA, Cdc42, Rac1 ↑ RhoB, TIMP-2	[236,237,238,239]
Rosmarinic acid	Decrease of tumor cells viability, induction of apoptosis, inhibition of proliferation, invasion and migration ↓ expression of NLRP3 inflammasome, ADAM17, EGF, phospsho-AKT, phospho-GSK3β, MMP-2, MMP-9, Bcl-2 ↑ caspase -3, -8, Bax	[240,241,242,243]
Syringic acid	Inhibition of UVB-induced carcinogenesis ↓ COX-2, MMP-1, MMP-13, prostaglandin E2, phospho-EGFR, phospho-Erk1/2, phospho-JNK1/2, phospho-p38 MAPK, phospho-MEK1/2, phospho-MKK4/7, phospho-MKK3/6, phospho-B-Raf, phospho-Akt, phospho-Src	[248]

**Table 3 molecules-28-06251-t003:** Mechanisms of antiproliferative effect of flavonoids based on *in vitro* studies. Arrows indicate an increase (↑) or decrease (↓) in the levels/activity of the molecules.

Flavonoids	Mechanism of Action	References
Apigenin	Reduction of viability, migration, invasion, angiogenesis and inflammation, induction of apoptosis, S and G2/M phase cell cycle arrest, reduction of Ki-67 positive cells ↓ phospho-Erk1/2, phospho-Akt, Akt, JNK, phospho-mTOR, phospho-STAT3, MMP-2, MMP-9, VEGF, Twist1, PD-L1, phospho-Src, phospho-JAK2, N-cadherin, fibronectin, Bcl-2, IL-6, MCP-1, IL-12, IL-10, COX2, Cdk2, phospho-p70S6K, phospho-p90RSK, phospho-MSK, phospho-4E-BP1, phospho-AMPK, phospho-CREB, USF-2, phospho-MEK1/2, phospho-MKK4, phospho-MKK3/6, phospho-EGFR, phospho-c-Raf ↑ E-cadherin, keratin-8, Bax, cleaved PARP, cleaved caspase -9, p53, LC3-II	[251,252,253,254,255,256,257,258]
Astragalin	Induction of DNA fragmentation and apoptosis, decrease of viability and G2/M cell cycle arrest, modulation of immune response ↓ cyclin D1, Mcl-1, SOX10, procaspase -9 ↑ Bax, cleaved caspase -3, cleaved PARP	[262,263]
Catechin	Inhibition of cell proliferation, induction of apoptosis and cell cycle arrest, inhibition of autophagy, decrease of Ki-67 positive cells ↓ Bcl-2, Beclin-1, LC3-I/II, phospho-AMPK, PD-L1/PD-L2 expression, phospsho-STAT3, STAT, IRF1, Sirt3 ↑ caspase -3 activation, phospho-PI3K, phospho-Akt, phospho-mTOR	[267,268]
Chrysin	Inhibited of cell proliferation, migration and invasion, G2/M cell cycle arrest and induction of apoptosis, modulation of immune response ↓ MMP-2, MMP-9, VEGF, PI3K, phospho-Akt, PKC, phospho-FAK, RhoA, phospho-c-Jun, N-cadherin, GRB2, Ras, NF-кB, phospho-Erk1/2, cytosolic Bax ↑ E-cadherin, caspase-3 activation, PBG-D, phospho-p38, mitochondial Bax	[269,270,271]
Diosmin	Induction of apoptosis, ROS generation and DNA fragmentation, inhibition of invasion, migration and cell growth ↓ Bcl-2, MMP-2, MMP-9 ↑ p53, caspase -3, -9	[273,274,275]
Galangin	Inhibition of colony formation, cell motility, adhesion and migration, reduction of Ki-67 positive cells, induction of apoptosis and autophagy ↓ phospho-FAK, FAK, phospho-Akt, phospho-Erk1/2 ↑ GCLC, GSS, phopsho-Nrf2, Nrf2	[276,277,279]
Genistein	Inhibition of cell proliferation, growth, viability and migration, S and G2/M phase cell cycle arrest, ROS generation and induction of ER stress-mediated apoptosis ↓ Cdk1, phospho-FAK, phospho-paxilin, phospho-p38, phospho-Erk1/2, phosho-JNK, tensin-2, vinculin, α-actinin, cyclin E and B, Cdc25C, Bcl-2, Bcl-xL ↑ activation of Chk2, p21, p27, p53, APAF	[207,283,284,285,286,287,288,289]
Isoquercitrin	Inhibition of viability and clonogenicity, DNA fragmentation, G1/S cell cycle arrest, induction of apoptosis ↓ Bcl-2, procaspase -8, -9, phospho-PI3K, phospho-Akt, phospho-mTOR ↑ Bax, AIF, Endo G, cleaved PARP	[297]
Isorhamnetin	Induction of apoptosis, inhibition of cell proliferation, growth, migration and colony formation, G2/M cell cycle arrest ↓ phospho-Akt, nuclear NF-кB, COX-2, phospho-Erk1/2, Bcl-2, COX-2, phosphpo-MEK1/2, phospho-Erk1/2, phospho-p90RSK ↑ activation of caspase -3, Bax	[299,300]
Kaempferol	Inhibition of cell proliferation, growth, migration and invasion, induction of apoptosis, G2/M cell cycle arrest ↓ phospho-PI3K, phospho-Akt, phospho-mTOR, activity of hexokinase, VDAC1, Akt/GSK-3β pathway, COX-2, phospho-Erk1/2, phospho-p38, phospho-JNK, Src activity	[303,304,305,306]
Luteolin	Suppression of cell proliferation, migration and invasion, induction of DNA damage and apoptosis, ER stress and ROS generation, G0/G1 and G2/M cell cycle arrest ↓ phospho-Akt, phospho-PI3K, MMP-2, MMP-9, HIF-1α, VEGF-A, VEGFR-2, MMP-2, MMP-9, N-cadherin, vimentin, integrin β3, STAT3, ZEB1, phospho-FAK, phospho-Src, Src ↑ TIMP-1, TIMP-2, E-cadherin, PERK, eIF-2α, ATF6, CHOP, cleaved caspase -12	[308,310,311,312,313,314]
Myricetin	Inhibition of cell proliferation, colony formation, invasion, migration, inflammation and angiogenesis, induction of apoptosis and ROS generation ↓ COX-2, Fin kinase activity, Bcl-2, MMP, NF-кB, phospho-Erk1/2, phospho-38, phospho-JNK, phospho-p90RSK, phospho-MSK, phospho-MEK, MEK1, phospho- Raf, phospho-JAK1, AP-1, c-Fos, STAT3, ↑ Bax, p53, GADD45, caspase -3, -8, -9	[319,320,321,322,323,324]
Naringenin	Induction of apoptosis, DNA fragmentation, ROS generation and mitochondrial damage, suppression of proliferation, migration, metastasis, tube formation and angiogenesis ↓ phospho-Erk1/2, phospho-JNK, Tie2, TPC2 activity, ↑ caspase -3 activation, cleavage of PARP, transglutaminase activity	[325,326,328,329]
Quercetin	Reduction of cell viability, proliferation, colony formation, migration and invasion, induction of DNA fragmentation and ROS generation, cell cycle arrest, inhibition of EMT ↓ MMP, Bcl-2, PKC-α, STAT3 expression, phosphorylation and nuclear localization, phospho-Src, phospho-JAK2, Mcl-1, Bcl-xL, MMP-2, MMP-9, VEGF, PKC, N-cadherin, vimentin, fibronectin, Twist, Snail, S100A7, phospho-Met, FAS, phospho-Fak, Fak, Gab1, phospho-Pak1, Nrf-2, catalase, B-Raf, phospho-MEK, phospho-Erk1/2, phospho-Akt, PI3K ↑ phospho-JNK, phospho-p38, E-cadherin, VCAM1, ICAM1, cleaved PARP, Bax, cleaved caspase -3, -8, Bim, AIF, RIG-I, STAT1, IRF7	[334,336,337,338,339,340,341,342,343,345,346]
Rutin	Decrease of cell viability, nuclear fragmentation, induction of apoptosis ↓ COX-2, iNOS, phospho-STAT3, AP-1, phospho-JNK ↑ β-galactosidase, p38 MAPK	[353,354,355]
Vitexin	Induction of apoptosis, DNA damage, ROS generation and oxidative stress G2/M cell phase cell cycle arrest, inhibition of colony formation ↓ Bcl-2, Cdk1, Cdk6, cyclins A2 and E2, MMP-2, MMP-9, vimentin, Slug, Twist, phospho-Src, phospho-JAK1/2, Chk2 ↑ Bax, cleaved PARP, phospho-ATM, phospho-ATR, GADD45, p21, PUMA, phospho-Chk2, p53, γ-H2AX	[356,357]

**Table 4 molecules-28-06251-t004:** Mechanisms of antiproliferative effect of other phenolic compounds based on *in vitro* studies. Arrows indicate an increase (↑) or decrease (↓) in the levels/activity of the molecules.

Other Phenolic Compounds	Mechanism of Action	References
1′-acetoxychavicol acetate	Inhibition of ROS production and lipid peroxidation ↓ NF-кB	[358,359]
*trans*-Anethole	Inhibition of cell proliferation, colony formation, induction of apoptosis, modulation of MiR-498/STAT4 axis ↑ miR-498	[360,363]
Arbutin	Induction of mitochondrial dysfunction ↓ MMP, Bcl-2, Bxl-xL, vimentin, HSP90, α-enolase, Inosine-5′-monophosphate dehydrogenase 2, Peroxiredoxin-1 ↑ p53, VDAC-1, 14-3-3G	[365,366]
Capsaicin	Inhibition of cell growth, proliferation, migration, induction of intrinsic and extrinsic apoptosis and autophagy and DNA damage, ROS and RNS generation ↓ phospho-PI3K, phospho-Akt, phospho-mTOR, Rac1, Bcl-2, tNOX, MDM2, procaspase -3, -8, -9, SOX2, EZH2, Sirt1 ↑ p53 activation, Bax, DR4, Fas, PARP cleavage, caspase -3 activation, ATG5, ATG7, Beclin-1, LC3-I/II, iNOS, cytochrome *c* release,	[368,369,370,371,372,373]
Carnosic acid	Inhibition of growth, proliferation, migration, adhesion and colony formation, G0/G1 cell cycle arrest and induction of apoptosis ↓ p27, MMP-2, MMP-9, TIMP-1, uPA, VCAM-1, N-cadherin, vimentin, Snail, Slug, phospho-Src, phospho-Fak, phospho-Akt ↑ p21, E-cadherin, TIMP-2	[375,376,377]
Carnosol	Inhibition of cell growth, proliferation, invasiveness and colony formation, induction of ROS production, DNA damage and apoptosis ↓ MMP-2, MMP-9, phospho-Erk1/2, phospho-p38, phospho-JNK, phospho-Akt, NF-кB and c-Jun nuclear translocation, phospho-Src, phospho-STAT3, cyclin D1, D2, D3, survivin, phospho-γH2AX, phospho-Chk1, iNOS, COX-2, TNF-α, IL-1β, TIMP-2, Bcl-2, Bcl-xL, MDM2, ↑ p53, cleaved caspase -3, -7, -9, cleaved PARP, Bax	[381,382,385]
Carvacrol	Suppression of cell proliferation, growth, induction of apoptosis and necroptosis, G2/M cell cycle arrest, ROS production, DNA fragmentation and mitochondrial damage ↓ Bcl-2, TOMM20, procaspase -3, ↑ cleavage of PARP, Bax, cytochrome *c*, p53, γH2AX	[390,392]
Cinnamaldehyde	Inhibition of tumor cell proliferation, migration, invasion and angiogenesis, G1 phase cell cycle arrest, induction of ROS production and apoptosis ↓ HIF-α, VEGF, PI3K, phospho-Akt, phospho-mTOR, MMP-2, MMP-9, IL-8, phospho-Flk1, CD31, vimentin, Zeb1, Twist ↑ caspase -3 activation, HO-1, p21,	[396,397,398]
Curcumin	Inhibition of cell survival, viability, proliferation, migration, invasion, induction of ROS generation, DNA damage, oxidative stress, ER stress, apoptosis and autophagy, G2/M cell cycle arrest ↓ MMP-2, MMP-9, phospho-JAK2, phospho-STAT3, activity of NF-кB, phospho-Erk1/2, iNOS, phospho-IκBα, p38, COX-2, PGE2, NO, cyclin D1, Bcl-2, Mcl-1, MMP, DNA-PKcs, XIAP, phospho-Akt, phospho-mTOR, phospho-P70S6K, phospho-S6, phospho-4EBP1, phospho-Rb, G6PD, GSH ↑ Bax, activation of caspases -3, -7, -8, -9, -12, TIMP2, phospho-p53, p53, p21, p27, Chk2, γH2AX, cleaved PARP, Fas, phospho-JNK, HLJ1, GRP94, GRP78, phospho-EIF2α, GADD153, cytochrome *c* release, HIF-1α, Bim1, Foxo3 nuclear translocation, MST1	[399,400,401,402,403,404,405,406,407,408,410,411,412,414]
Ellagic acid	Induction of ROS generation and apoptosis, suppression of proliferation, migration and invasion, G1 phase cell cycle arrest ↓ NF-кB, phospho-EGFR, β-catenin, Keap1, vimentin, IL-1, IL-8, TNF-α, ↑ HO-1, SOD, Nrf2, IL-10, E-cadherin,	[415,416,417,418]
Eugenol	Inhibition of tumor cell proliferation and colony formation, induction of DNA fragmentation, apoptosis, G1 and G2/M phase cell cycle arrest ↓ MMP, cyclin A, D3, E, Cdk2, Cdk4, cdc2, E2F1, c-Myc, H-Ras, Bcl-2, ODC activity, COX-2, iNOS, IL-6, TNF-α, PGE2, NF-κB/p65 translocation ↑ activation of caspase -3, -6, -7 and lamin A, PARP and DFF45 cleavage, AIF and cytochrome *c* release, Bax, p53, p21,	[423,424,425,426,428,430,431]
6-gingerol	G1 cell cycle arrest, ROS generation and induction of apoptosis, modulation of immune response, inhibition of angiogenesis ↓ MMP, Bcl-2, COX-2, NF-κB activation, cyclin D1, E, A, phospho-p38, phospho-IκBα, phospho-ATF2, LPO ↑ Bax, caspase -3, -9, cytochrome *c*, Apaf-1,	[436,437,438,439,440]
6-shogaol	Inhibition of cell survival, proliferation, induction ↓ iNOS, COX-2, NF-κB nuclear translocation, phospho-IκBα, phospho-Erk1/2, phospho-p38 MAPK, phospho-PI3K, phospho-Akt, phospho-JNK1/2, IL-6, IL-10, TNF-α, HO-1 ↑ Nrf2	[443,444]
Thymol	Reduction of UVA and UVB-induced DNA damage and oxidative stress	[447,448]

## Data Availability

Not applicable.

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
