# Peer review of "Spice-Derived Phenolic Compounds: Potential for Skin Cancer Prevention and Therapy"

_molecules, 2023, doi:10.3390/molecules28176251_

Round 1

Reviewer 1 Report

This review focuses on the potential of spice-derived phenolic compounds as preventive or therapeutic agents for managing skin cancers.

By compiling and analyzing the available knowledge, the authors provide an excellent and complete review that can guide future research in identifying new anticancer phytochemicals and uncovering additional mechanisms for combating melanoma and non-melanoma skin cancers.

This review provides many details about types of skin cancer, an overview of spices, phenolic compounds present in these compounds, amount others.

Given the large amount of information that is detailed, it would be desirable to add some tables summarizing the information provided. I suggest incorporating tables detailing the phytochemicals and phenolic compounds of selected spices or their pharmacological activities. Also, the authors must include tables summarizing the effect of spices-derived phenolic compounds against melanoma and non-melanoma skin cancer, emphasizing signaling pathways and other molecular mechanisms (effect on cell cycle progression or cell death, cell metabolism, among others).

Reviewer 2 Report

The study focused to review  phenolic compounds: potential for skin cancer prevention and therapy. These findings are very impressive, however, lack of in vivo studies and the underlying mechanisms required to be addressed.

Round 2

Reviewer 2 Report

This is accepted to publish

Minor editing of English language required